# Molecular basis of foreign DNA recognition by BREX anti-phage immunity system

Alena Drobiazko [1,7], Myfanwy C. Adams[2,7], Mikhail Skutel [1], Kristina Potekhina[1], Oksana Kotovskaya[1], Anna Trofimova[1,3], Mikhail Matlashov[1], Daria Yatselenko [1], Karen L. Maxwell [4], Tim R. Blower [5], Konstantin Severinov[3,6] ✉, Dmitry Ghilarov [2] ✉ & Artem Isaev [1] ✉

Anti-phage systems of the BREX (BacteRiophage EXclusion) superfamily rely on site-specific epigenetic DNA methylation to discriminate between the host and invading DNA. We demonstrate that in Type I BREX systems, defense and methylation require BREX site DNA binding by the BrxX (PglX) methyl-transferase employing S-adenosyl methionine as a cofactor. We determined 2.2-Å cryoEM structure of *Escherichia coli* BrxX bound to target dsDNA revealing molecular details of BREX DNA recognition. Structure-guided engineering of BrxX expands its DNA specificity and dramatically enhances phage defense. We show that BrxX alone does not methylate DNA, and BREX activity requires an assembly of a supramolecular BrxBCXZ immune complex. Finally, we present a cryoEM structure of BrxX bound to a phage-encoded inhibitor Ocr that sequesters BrxX in an inactive dimeric form. We propose that BrxX-mediated foreign DNA sensing is a necessary first step in activation of BREX defense.

Bacteriophages (phages) are the most abundant biological entitiy on Earth shaping the structure and evolution of microbial communities[1,2]. To withstand phage infection, bacteria developed an arsenal of immunity systems, the diversity of which have been appreciated only in recent years[3–6]. Anti-phage defense strategies can be broadly classified into three main groups: avoidance of host cells recognition by phages by limiting the availability of surface receptors[7]; destruction of phage genomes or inhibition of essential phage processes[4]; abortive responses that inhibit the metabolism of or kill an infected cell, thus halting phage propagation in the population[8,9]. The function of host innate immunity systems depends on the ability to discriminate between self (host) and non-self (virus) molecules. A range of defense systems targeting phage DNA rely on epigenetic modifications, such as DNA methylation, as a marker of self[10]; a classic example being restriction-modification (R-M) systems present in ~83% of sequenced prokaryotic genomes[11]. A multitude of recently described immunity

systems, such as Dnd[12], Ssp[13], Dpd[14], DISARM[15], MADS[16], and BacteR-iophage EXclusion (BREX)[17,18] also encode DNA modification proteins. The mechanisms of phage defense by these systems are not fully determined and likely are more sophisticated than direct nucleolytic cleavage of non-modified invading DNA[19,20].

BREX antiviral immunity systems are found in ~7% of prokaryotic genomes, often clustered together with other immunity-related genes in defense islands[11,17,21]. Based on protein composition, BREX systems are classified into six types[17], with Type II BREX previously described as a Pgl (phage growth limitation) system of *Streptomyces*[22,23]. The most widespread Type I BREX employs an R-M like principle: it modifies cell's own DNA at specific sites (BREX sites), and this epigenetic mark protects host DNA from BREX activity. Invading foreign DNA does not have the modification and is sensitive to BREX defense. The acquisition of BREX-specific modification is sufficient to protect infecting phage from exclusion[18]. BREX methylation is strand-specific, i.e., only one

[1]Skolkovo Institute of Science and Technology, Moscow, Russia. [2]Department of Molecular Microbiology, John Innes Centre, Norwich, UK. [3]Institute of Gene Biology, Russian Academy of Sciences, Moscow, Russia. [4]Department of Biochemistry, University of Toronto, Toronto, ON, Canada. [5]Department of Biosciences, Durham University, Durham, UK. [6]Waksman Institute of Microbiology, Piscataway, NJ, USA. [7]These authors contributed equally: Alena Drobiazko, Myfanwy C. Adams. ✉e-mail: severik@waksman.rutgers.edu; Dmitry.Ghilarov@jic.ac.uk; artem.isaev@skoltech.ru

strand of an asymmetric recognition site (**GGTA<u>A</u>G** in *E. coli* HS) is methylated. Similar methylation patterns are employed by Type ISP, Type IIL, and Type III R-M systems[24–26]. As new non-methylated sites in host DNA will constantly emerge after each replication cycle, the activity of these defense systems needs to be tightly regulated to avoid targeting of host DNA. For example, Type ISP and III R-M systems require simultaneous recognition of two non-methylated sites on opposite DNA strands, followed by translocation of one (Type III) or both (Type ISP) restriction complexes along the DNA and their subsequent collision to licence DNA cleavage[27,28]. Type IIL restriction enzymes assemble into multimeric complexes whilst remaining bound to their cognate site, and the probability of restriction is determined by the local density of non-methylated sites[29]. How BREX systems recognize foreign DNA and avoid self-toxicity associated with emerging non-methylated sites in replicating host DNA is currently unknown.

With the exception of Type IV, all BREX systems encode BrxX (PglX) N6-adenine-methyltransferases[17,18,30–34] thought to be responsible for DNA modification[18]. Other core BREX proteins include BrxZ (PglZ) belonging to the alkaline phosphatase superfamily and homologous to the phosphodiesterase PorX[35], and BrxC (PglY), an AAA+ ATPase with distant homology to ORC/Cdc6 (Origin Recognition Complex) proteins[36,37]. In addition, Type I BREX systems encode a DNA-binding AAA+ ATPase BrxL[38], and two small proteins of unknown function BrxA[39] and BrxB (Fig. 1A). A model explaining BREX-mediated exclusion of phage DNA is currently lacking. BREX likely acts at the early stages of infection as it prevents phage DNA synthesis[17,18]. While BREX methylates host DNA similarly to R-M systems, it lacks an obvious restriction component. Yet, BREX is inhibited by viral anti-defense proteins that inhibit R-M systems, most notably, a DNA mimic Ocr that interacts with BrxX[40,41]. BREX methylation is also inhibited by the depletion of the intracellular S-adenosyl methionine (SAM) pool by phage T3 SAM lyase. Strikingly, SAM depletion also inhibits BREX defense, implying that like in some Type I R-M systems, SAM could perform regulatory functions in addition to being a donor of methyl groups[42,43].

In this work, we demonstrate that BREX response requires specific recognition of non-methylated BREX sites by methyltransferase BrxX, and that BrxX binding is enhanced by SAM. A 2.2 Å cryogenic electron microscopy (cryoEM) structure of a BrxX-DNA complex shows that both target recognition and methyltransferase domains (MTDs) are involved in sequence-specific recognition of BREX sites. Using cryoEM we further demonstrate how DNA mimic anti-restriction protein Ocr of bacteriophage T7 prevents DNA binding by BrxX and locks it in an inactive dimeric state. Finally, we establish that both BREX defense and methylation require assembly of a multi-subunit BrxBCXZ complex, allowing us to discuss potential models of BREX-mediated phage exclusion.

## Results

### BrxX mediates site-specific DNA recognition by the BREX system

We sought to determine at which stage of infection is BREX defense activated. BREX does not affect phage adsorption but prevents establishment of injected λ DNA in live fluorescence microscopy experiments, suggesting that viral DNA is degraded, occluded by BREX proteins or that injection itself is blocked[18]. Given that injection of λ DNA into host cells triggers ion efflux[44], we carried out the potassium efflux assay to monitor λ genome injection[45]. We measured extracellular K+ concentration in BREX− and BREX+ cultures, as well as in cultures of cells harboring the Type II R-M EcoRV system upon high MOI λ infection (Fig. 1B). A ΔlamB strain lacking the λ receptor was used as a negative control. The initial efflux dynamic was identical for all cultures, however, the BREX+ and EcoRV+ cells eventually restored the K+ potential, confirming that infection had failed. The results suggest that BREX does not interfere with the phage genome injection process, thus BREX defense must be initiated by the recognition of phage

infection-associated molecular signatures once they appear in the cytoplasm.

Type I BREX of *E. coli* strain HS methylates the fifth adenine in asymmetric **GGTA<u>A</u>G** motifs in host DNA; λ phage propagated in BREX + strains and carrying this BREX-specific methylation is insensitive to BREX[18]. This implies that BREX defense requires the recognition of non-methylated BREX sites in infecting phage genomes. To identify component(s) of the BREX system that are responsible for site-specific recognition of non-methylated DNA, we purified each of the six *E. coli* HS BREX proteins from cells transformed with appropriate expression plasmids and carried out electrophoretic mobility gel shift assays (EMSA) with a 43 bp dsDNA probe carrying a single unmodified BREX site. A probe with a scrambled BREX site was used as a control (Fig. 1C, Supplementary Fig. S1A). Consistent with previous observations, BrxA efficiently bound both probes[39]. BrxL also bound both specific and non-specific DNA although with affinity that was lower than previously reported[38], possibly suggesting that the 43 bp substrate used is too short for efficient binding. Importantly, only BrxX showed enhanced binding to the BREX site-containing probe. This suggests that BrxX is responsible for sensing non-methylated DNA essential for (at least) two core BREX functions: BREX defense activation if non-methylated DNA is foreign, and DNA methylation to protect it from BREX if non-methylated DNA belongs to the host. Deletion of *brxX* does not result in toxicity[18], as would be expected if BrxX had only methylation function and the remaining components of the BREX system were recognizing and attacking non-methylated sites in host DNA. This is consistent with a direct role of BrxX in the initiation of BREX defense, or at least in the assembly of the defense-competent complex. To further demonstrate the role of BrxX as a sensor of non-methylated BREX sites, we sequenced DNA cross-linked to BrxX in BREX+ culture infected with T7 phage ("Strep-Seq" procedure, see "Methods"). Mapping of reads on phage DNA confirmed the presence of peaks centered around BREX sites on both DNA strands (Fig. 1D) and demonstrated the expected strand cross-correlation despite the relatively low amplitude of the signal (Fig. 1E). The result shows that BrxX binds non-methylated BREX sites in phage DNA in vivo, the likely first step in the activation of BREX defense.

### SAM and methylation state modulate the affinity of BrxX to BREX sites

S-adenosyl methionine (SAM) serves as a donor of methyl groups for methyltransferases. SAM also can stabilize DNA-bound protein complexes[42,46], and was recently shown to be required for BREX defense[43]. We studied how the presence of SAM affects DNA binding by BrxX (Fig. 1C). BrxX non-specifically bound the 43 bp dsDNA probe with low affinity. The binding was improved ~3-fold in the presence of SAM ($K_d^{SAM+}$ ~300 nM compared to $K_d^{SAM−}$ >1 μM). The enhancing effect of SAM was ~5-fold when the substrate contained a BREX site ($K_d^{SAM+}$ ~60 nM compared to $K_d^{SAM−}$ ~300 nM), suggesting that SAM binding promotes BREX sites recognition or results in more stable BrxX:DNA complexes. To further study BrxX substrate preferences we performed EMSAs with BREX site-containing dsDNA fragments of different lengths. Decreasing dsDNA length to 20 or 30 bp resulted in weaker binding while extending the length to 50 bp improved binding. The presence of a super-shifted band at higher BrxX concentrations suggested non-specific binding of additional BrxX molecules to the same DNA fragments (Supplementary Fig. S1B). BrxX also efficiently bound ssDNA substrates, although without a clear preference for the presence of BREX sites, suggesting that bases from both the top and bottom DNA strands are required for recognition (Supplementary Fig. S1C). To further quantify BrxX interaction with dsDNA, we determined the $K_d$ values by an orthogonal approach using biolayer interferometry (BLI) with a biotinylated 40 bp dsDNA probe carrying a single BREX site or the polyC sequence (Supplementary Fig. S2A). Although BLI confirmed BrxX preference for DNA containing BREX site

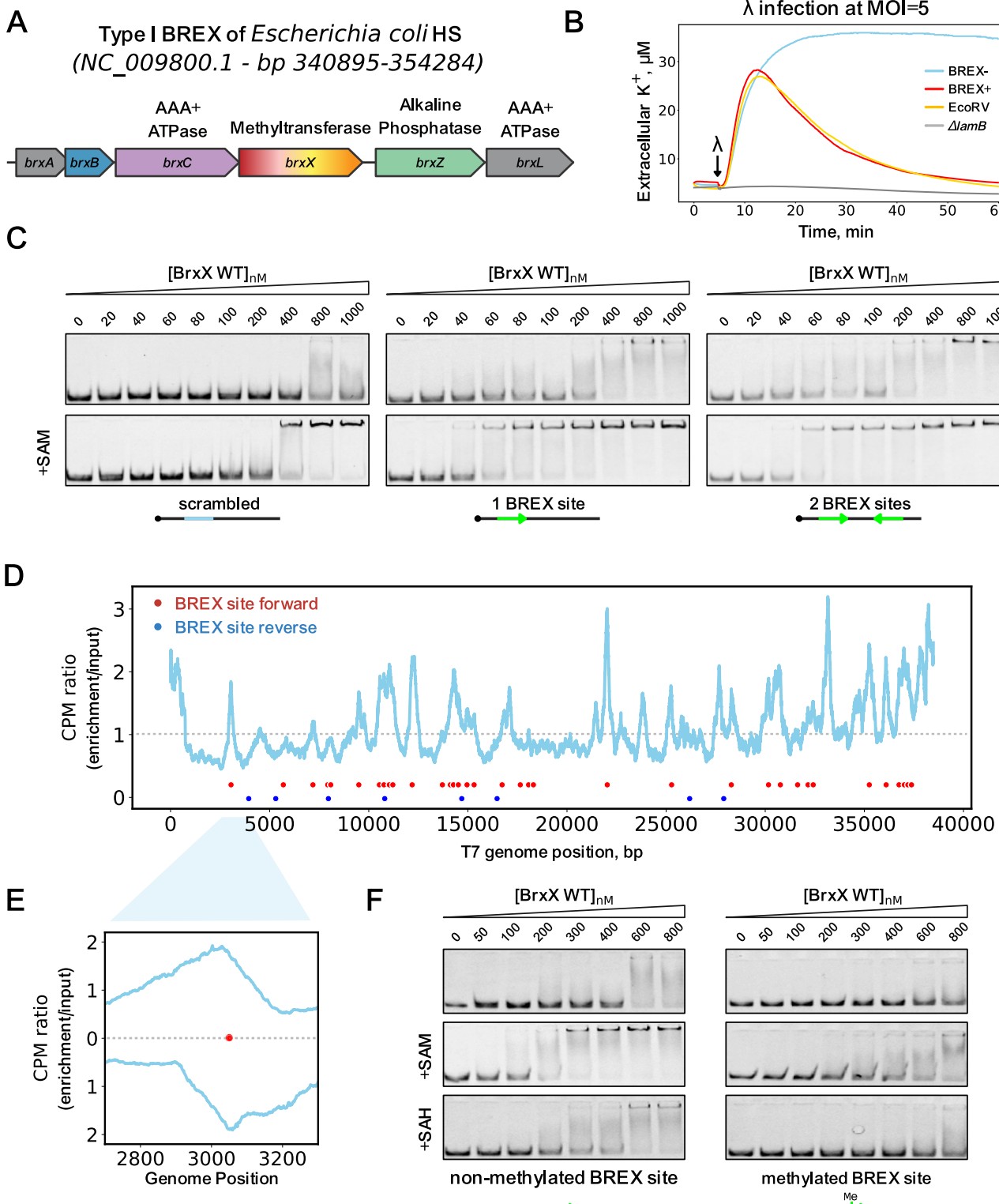

**Fig. 1 | The BrxX (PglX) methyltransferase specifically recognizes BREX sites in DNA. A** The Type I BREX locus of *E. coli* HS, its genomic position, and predicted functions of encoded proteins. **B** Potassium efflux assay with *E. coli* BW25113 BREX⁻, BREX⁺, EcoRV, or ΔlamB cultures infected with phage λ at MOI = 5. A representative result from three independent experiments is shown. **C** EMSA with 20 nM Cy5-labeled 43 bp dsDNA substrates with no BREX sites (scrambled), or with one or two BREX sites, incubated with an indicated amount of BrxX in the presence or in the absence of 0.5 mM SAM. Green arrows indicate the orientation of BREX sites. Representative gels from experiments performed in triplicate are shown. **D** Strep-seq analysis of DNA cross-linked to BrxX-Strep 15 min after infection of a BREX⁺

culture with T7_fusion phage at MOI = 1. Reads from the enriched sample were mapped to T7 genome and the signal was normalized to the bulk DNA level and genome size to obtain CPM values. Positions of BREX sites (**GGTA<u>A</u>G**) on the top and bottom DNA strands are indicated with red and blue dots, correspondingly. **E** Cross-correlation analysis (normalized coverage of the forward and reverse reads) for a representative peak in the T7 genome. **F** EMSA with 20 nM of 40 bp Cy5-labeled dsDNA substrate with methylated or non-methylated BREX site incubated with indicated amounts of BrxX. Representative gels from triplicate experiments are shown.

in the presence of SAM, accurate $K_d$ estimation was not possible due to heterogenous binding likely resulting from the interaction of more than one BrxX protein with the dsDNA substrate (Supplementary Fig. S2B, D). To overcome this, we measured $K_d$ using a shorter 20 bp dsDNA substrate, which showed weaker binding but fit the 1:1 interaction model. The $K_d$ values obtained confirm the stimulating effect of SAM and the preference for BREX site-containing DNA, in agreement with the EMSA results (Supplementary Fig. S2C, E).

Reduced affinity to methylated substrates is a common mechanism of control of methyltransferases activity in R-M systems[47,48]. At a post-methylation stage, the methylated adenine is accommodated in the enzyme catalytic pocket together with S-adenosyl homocysteine (SAH), another product of the methylation reaction. When SAM-bound methyltransferase binds methylated DNA, SAM interferes with the correct placement of the methylated adenine in the catalytic pocket. To determine if methylation affects the binding of BrxX to its site, we prepared a pair of identical dsDNA substrates with either methylated or non-methylated BREX site and compared their binding to BrxX in three conditions: without co-factors, in the presence of SAM (to mimic BrxX interaction with methylated DNA), or in the presence of SAH (to mimic the post-methylation BrxX:DNA complex) (Fig. 1F). As expected, BrxX had a much lower affinity for the methylated substrate under all conditions tested. This suggests that: (i) DNA methylation prevents recognition by BrxX; (ii) post-methylation complex is not stable, promoting the release of BrxX and, likely, subsequent exchange of SAH for SAM; and (iii) methylation sensing by BrxX does not strictly require the presence of a cofactor.

We further checked whether BrxX interacts with BREX-methylated DNA of phage λ or with glucosylated DNA of phage T4 since both modifications provide full resistance against BREX defense in vivo[18] (Supplementary Fig. S3A, B). As controls, we used non-methylated λ gDNA or non-glucosylated T4$_{147}$ gDNA, that only retains the 5-hydroxymethylcytosine modification[49] (Supplementary Fig. S3C, D). BrxX bound both modified and non-modified DNA substrates. It is not possible to accurately compare binding affinities for such large DNA molecules, however, the results indicate that methylation or glucosylation of DNA does not prevent non-specific binding by BrxX and instead affects BREX site recognition or downstream reactions.

**Overall architecture of the BrxX-DNA complex**

We sought to understand the molecular mechanism for site-specific recognition and subsequent methylation of DNA by BrxX. We failed to detect DNA-bound species following cryoEM analysis of BrxX in the presence of 14 and 20 bp DNA oligonucleotides containing a single BREX site. However, when BrxX was incubated with a 43 bp dsDNA fragment containing two BREX sites in a head-to-tail (H2T) arrangement (DNA H2T; Fig. 2A) in the presence of SAM, a DNA-bound complex accounted for the majority of 2D class averages (Fig. 2C, Supplementary Fig. S4B). Single-particle analysis of 11,152 movie dataset demonstrated the formation of a 1:1 complex (**BrxX:DNA**) that was highly homogenous despite the presence of two BREX sites (see Supplementary Fig. S4A, B). Only the first (from the 5′-end) binding site was occupied (Fig. 2A), suggesting that a stretch of DNA downstream of a BREX site is essential for BrxX binding. Despite the relatively small protein size (139 kDa), analysis of **BrxX:DNA** resulted in a 2.2 Å global resolution map (Fig. 2B, C) with local resolution for areas around the six-base pair recognition sequence extending to 2.1 Å (see Supplementary Figs. S4C–E and S5A, B). This high-resolution data allowed for unambiguous identification and positioning of 25 out of 43 base pairs of DNA and clear side-chain modeling of BrxX core residues (Fig. 2C), whereas the remaining loosely bound DNA bases and peripheral loops in BrxX were observed at lower resolution (transparent contour in Fig. 2C; see also Supplementary Fig. S4D) due to their intrinsic flexibility.

The DNA-bound BrxX is a bi-lobed Pacman-like monomer consisting of four domains (Fig. 2B–D). The upper Pacman "jaw" contains

the N-terminal domain (NTD, 1-210) and the methyltransferase domian (MTD, 211-650) while the lower "jaw" comprises the target recognition domain (TRD, 651-837) followed by the C-terminal domain (CTD, 838-1205). The two "jaws" are connected by flexible hinge loops (Fig. 2D inset, residues 638−675) at the interface between the MTD and TRD. The NTD comprises an N-terminal α-helix followed by a three-stranded antiparallel β-sheet and a cluster of eight α-helices. The MTD displays a Rossman-like methyltransferase fold comprised of a nine-stranded β-sheet flanked by ten α-helices, and a type IV (NPPY motif)[50] catalytic pocket with bound SAM (Supplementary Figs. S5C, D, S6E). The TRD is responsible for site-specific DNA recognition and is folded into two α/β subdomains. The CTD consists of an α-helical part and a three-stranded antiparallel β-sheet, connected to a two-helix bundle (the lowermost part of Pacman's "jaw") by what appears to be a metal-binding site (Fig. 2D). A metal ion is coordinated by five residues: side chains of D997, D999, D1012, Y1113, and the main-chain oxygen of I1001, with the last valence occupied by a water molecule. These residues are highly conserved among BrxX homologs (Supplementary Fig. S6A, F). As the EM buffer contained $Mg^{2+}$, and because of the octahedral shape of the metal density and six-valence coordination, we hypothesize that the metal ion is $Mg^{2+}$, but its exact nature remains to be determined.

Conservation analysis of BrxX proteins closely related to *E. coli* HS BrxX by ConSurf server[51] identified the TRD as the most variable part of the protein, presumably reflecting the diversity of sequences recognized by different BREX systems. In contrast, sequences of CTD, particularly around the metal-binding site, and of MTD and NTD, particularly around the SAM-binding pocket, were highly conserved (Supplementary Fig. S6A, F). Yet, the NTD and CTD are not well conserved beyond Type I BREX systems (we failed to detect significant structural homology using DALI[52] or Foldseek[53] servers). Thus, these domains are unique for Type I systems and may have specific functions within a larger BREX complex(es) (see below). The BrxX MTD can be compared to corresponding domains of other DNA methyltransferases (Supplementary Fig. S6C, Supplementary Table S5): MmeI[54] and DrdV[29] (Type IIL R-M enzymes), and CamA[55] (an orphan methyltransferase). BrxX shares the highly conserved SAM-binding (motif I) and methylation (motif IV) motifs with these proteins[50] (Supplementary Fig. S6E). However, the BrxX MTD contains an insert between residues 390 and 500 that is not observed in other homologs and, alongside CTD and NTD, could participate in interactions with other BREX proteins.

The DNA bound to BrxX is bent by ~33° (Fig. 2E), with the most prominent distortion occurring around the six-base pair recognition sequence embedded between the MTD and TRD. The fifth adenine of the **GGTAAG** sequence is rotated out of the DNA backbone together with its deoxyribose moiety and the neighboring phosphodiester groups and is nestled in the active site of the MTD. There is a significant rotation of the opposite thymidine as well. The shifting of surrounding bases results in discontinuity of the non-target strand bases π-stacking. The major groove over the recognition sequence widens to ~25 Å while the minor groove widens to ~17 Å compared to 19 Å and 12 Å, respectively, in the B-form DNA. 29 out of 35 direct interactions between the BrxX monomer and DNA occur within the six bases involved in site-specific recognition. The distorted state of DNA required for site-specific recognition is stabilized by a series of positively charged residues forming direct contact with the phosphate backbone. A positively charged lining of the "upper jaw" consists of K226, K522, K515, and K533 of MTD (Figs. 2F and 3A); the complementary positively charged groove in the "bottom jaw" includes K719 and R944 of the TRD, and R1145 and K1152 of the CTD that stabilize the bent 3′ tail of the modeled DNA segment. This 3′ tail of the DNA fragment outside of the recognition site is essential for stable binding and lies within a positively charged groove of the BrxX-specific CTD (Fig. 2F). The overall shape and mode of DNA binding resembles the one observed in MmeI:DNA complexes (Supplementary Fig. S6D).

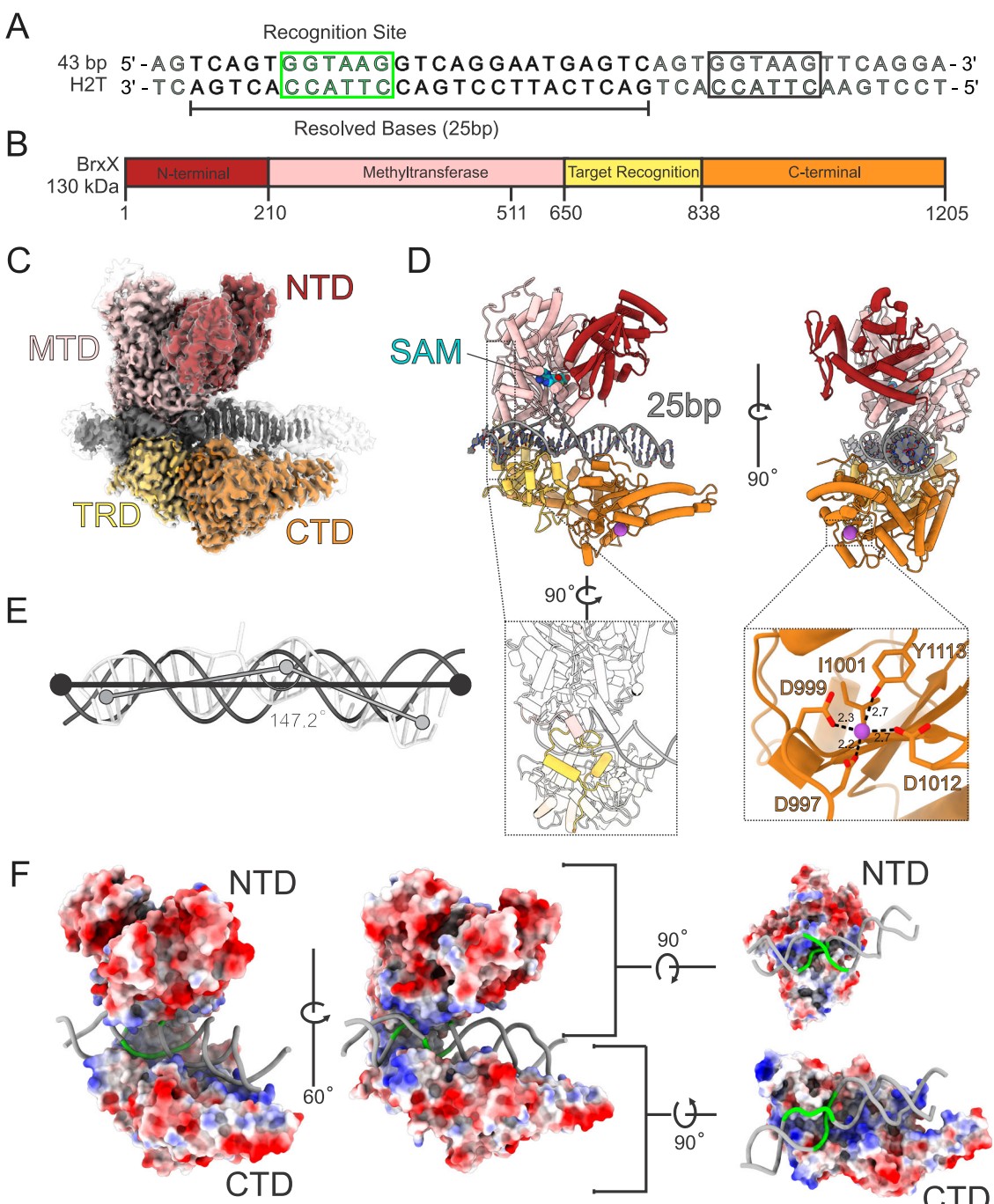

**Fig. 2 | Architecture of the BrxX-DNA complex. A** The 43 bp head-to-tail (H2T) DNA fragment used for cryoEM. The six-base pair BREX recognition sequence is shown in green, dark gray shows the resolved (modeled) bases whereas light gray shows unresolved bases. Black rectangle highlights the second (unbound) BREX site. **B** Domain arrangement of the BrxX monomer: N-terminal domain (NTD) is shown in dark red, methyltransferase domain (MTD) in pink, target recognition domain (TRD) in yellow, and C-terminal domain in orange (CTD). This color scheme is used throughout the manuscript. **C** CryoEM map, side view, of the BrxX-DNA complex. The sharpened map, contoured at 5σ, is colored according to the diagram above and shown within the contour of unsharpened map (also at 5σ, transparent contour). Only the density within the sharpened map was modeled, and resulting 25 bp of DNA was built. **D** A cartoon representation of an atomic model of the BrxX-DNA complex. SAM (cyan) and magnesium ion (purple) are shown as van der Waals sphere representations. The two inset regions highlight the flexible loops between the MTD and the TRD (i) and the metal-binding site in the CTD (ii). **E** A structural comparison of the 25 bp DNA bound to BrxX as seen in the cryoEM structure (gray tubes) to idealized B-form DNA (black cartoon). BrxX-bound DNA deviates from a linear trajectory by ~33° with widened major and minor grooves. **F** Surface coloring based on Coulombic electrostatic potential calculated by ChimeraX[76] where red and blue show, respectively, negative and positive potentials.

## Structural basis of site-specific DNA recognition by BrxX

All site-specific direct interactions with DNA bases happen within the major groove of DNA to allow base discrimination; they include both the target and non-target DNA strands (Fig. 3A, B). The TRD contributes to the recognition of the first four base pairs of the recognition

sequence. Specifically, R751, K681, and N823 recognize G1, G2, and A4 of the target sequence whereas D684 and Q784 recognize C2' and A3' of the non-target strand (Fig. 3A, B). The flipped adenine is flanked by an A:T base pair, which facilitates local DNA buckling and bending that accompanies base flipping. Involvement of residues from the MTD in

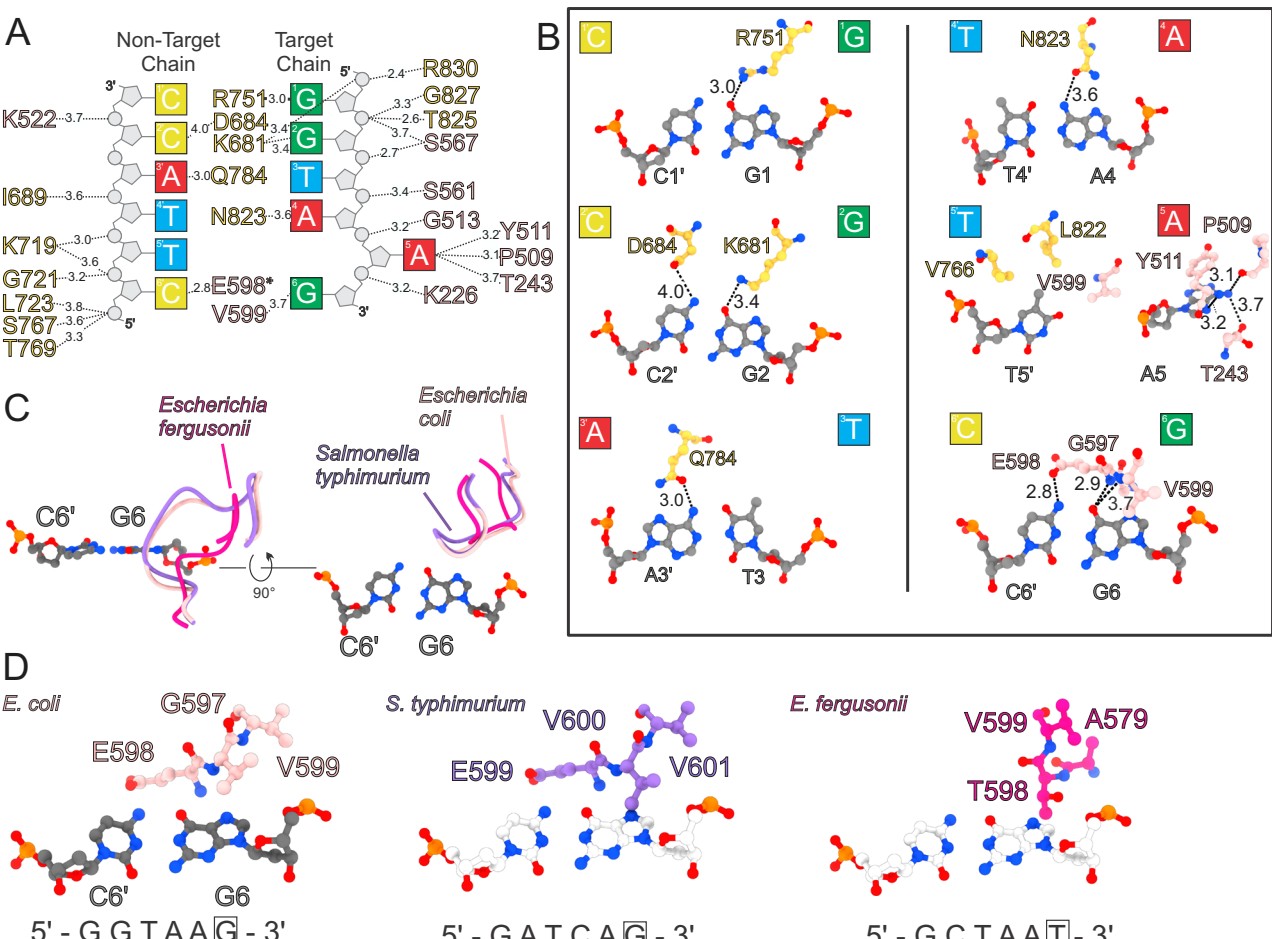

**Fig. 3 | Molecular basis of BREX site recognition by BrxX. A** A scheme of interactions between BrxX and the six-base pair BREX site. Distances in Angstrom are indicated. An asterisk indicates that the main chain and not the side chain is involved. Colors correspond to amino acids from MTD (pink) or TRD (yellow). **B** Molecular basis for site-specific recognition of the BREX site. Each base pair is presented separately and oriented with major groove on top. Distances in Angstrom are indicated. Non-target strand indicated by a prime symbol. **C** A species-specific MTD loop recognizes the sixth base pair. A structural superposition is shown of loop 587–601 of *E. coli* HS BrxX MTD (rose) and equivalent loops in

*Salmonella typhimurium* (purple) and *Escherichia fergusonii* (magenta) based on AlphaFold DB models AF-A0A5A8QD96-F1 and AF-B7L3T0-F1 respectively. **D** Side-by-side comparison of above loops with amino acids predicted to be important for the recognition of the sixth base pair shown as sticks. Recognition sites for all three BREX systems are shown, and the last base pair of the recognition site is boxed (G6 for *E. coli*). As the cognate DNA structure is not available for *S. typhimurium* and *E. fergusonii*, sixth base pair of *E.coli* HS BrxX is shown for all three models. In the case of *E. fergusonii*, a larger purine base would be replaced by a smaller pyrimidine, in line with an observed change in the MTD loop structure.

DNA recognition is not unusual among DNA methyltransferases[29,54,56], and in our structure occurs with the fifth and sixth positions of the BREX site. Adenine base flipping is facilitated by the intercalation of hydrophobic residues V766 and L822, that causes a 90° propeller-twist rotation of the opposite T5′ (see Fig. 3A, B), and V599 that sterically fills the empty space formed by the appearance of the extrahelical adenine. Together, these three residues create a hydrophobic patch to accommodate the C7 methyl group of T5′. Similar to other MTases, the flipped adenine is inserted deep into the methylation active site (see Fig. 2D; the adenine ring is stacked against Y511 and surrounded by hydrophobic residues L245, V557, F592, and I595. An amino group of the adenine forms direct interactions with T243 and N508 (3.6 Å and 4.5 Å, respectively), facilitating specific base recognition.

Our analysis identified a critical loop (residues 587–601) in *E. coli* BrxX MTD that confers specificity to the recognition of the final base pair of the BREX site. Not only does residue 598 make direct contacts with the target strand terminal base G6 (main-chain amide to O6, 2.9 Å) and the opposite C6′ (side-chain oxygen to N4, 2.8 Å; see Fig. 3C, D), but the loop sterically enables only the R:Y pairing in this position. The importance of this observation is revealed by comparison with the

BrxX homologs from *Salmonella typhimurium* and *Escherichia fergusonii* modeled by AlphaFold2. These BREX systems recognize, correspondingly, the **GATCAG** and **GCTAAT** sequences[31,32,57]. The loop in *S. typhimurium* BrxX is predicted to be very similar to the one in *E. coli* BrxX, as expected given the identity of the terminal base. In contrast, in *E. fergusonii* BrxX the equivalent loop is markedly shorter (see Fig. 3C, D), likely to accommodate the T:A base pair. We attempted to substitute *E.coli* BrxX loop with the one from *E. fergusonii* to manipulate the recognition specificity, however, three different hybrids we tested lost anti-phage defense (Supplementary Table S1).

## TRD engineering allows to change BREX site specificity and to enhance anti-phage defense

To validate the role of BrxX residues involved in non-specific and site-specific DNA recognition, we constructed a series of mutants and tested them in the efficiency of plating (EOP) assay (Fig. 4A, Supplementary Table S1, Supplementary Fig. S10). Mutations of the residues comprising positively charged lining of BrxX MTD (K515E/K522E/K526E/K533E) and CTD (K1076E/K1077E and K719E) presumably involved in non-specific DNA recognition abolished BREX defense

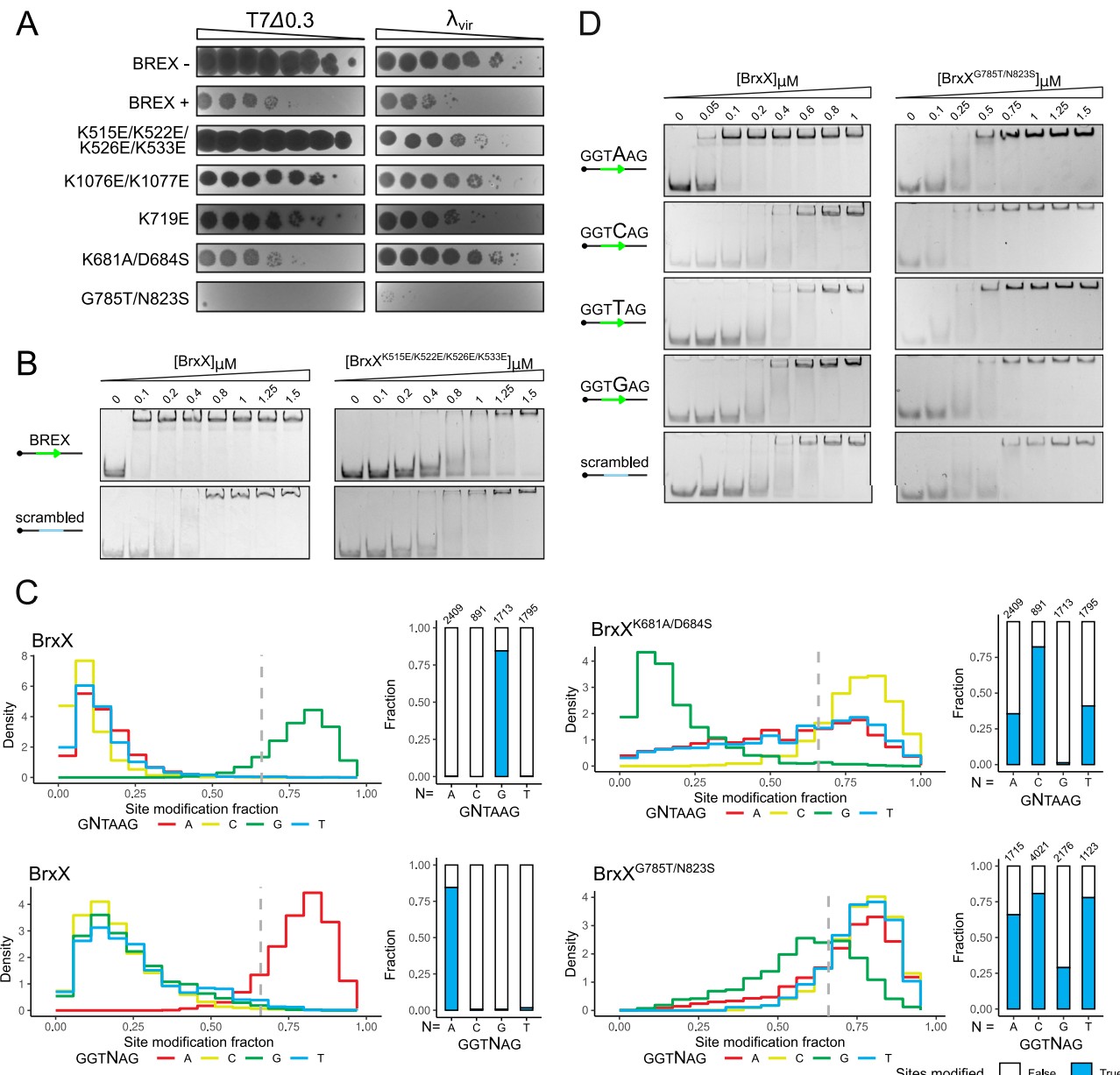

**Fig. 4 | Substitutions in BrxX TRD mediate BREX specificity change and enhance anti-phage defense. A** EOP assay with BREX-sensitive T7Δ0.3 and λ$_{vir}$ phages and cells carrying BREX systems encoding indicated BrxX mutants. **B** EMSA with 20 nM of 40 bp Cy5-labeled dsDNA substrate with intact or scrambled BREX site and WT or mutant BrxX proteins. **C** Substitutions in BrxX TRD result in relaxed BREX site specificity. Methylation of BREX sites in genomic DNA of BREX⁺ cells with WT BrxX or indicated BrxX mutants assessed through ONT sequencing. Diagrams

demonstrate the density distribution of modified sites for each possible nucleotide variation in the second (above) or fourth (below) position; a 66% threshold (counted as significant) is indicated with dotted lines (see "Methods"). The fraction of sites exceeding the modification threshold and the total number of available sites in the genome are provided in the insets. **D** EMSA with 20 nM of 40 bp Cy5-labeled dsDNA substrates with varying nucleotides at the fourth position of the BREX site or scrambled BREX site and WT or TRD mutated BrxX proteins.

(Fig. 4A, Supplementary Fig. S6B). EMSA carried with the MTD variant (BrxX$^{K515/K522E/K526E/K533E}$) demonstrated decreased binding and complete loss of site-specific DNA recognition, confirming the role of MTD in DNA stabilization by BrxX (Fig. 4B). To directly test the possibility of engineering BREX site specificity, we compared amino acids responsible for the recognition of the second and fourth nucleotides in *E. coli* BrxX with AlphaFold2-generated models of the *E. fergusonii* and *S. typhimurium* homologs with known DNA specificity (Supplementary Fig. S6G). In *E. coli* the G:C pair in the second position is recognized by K681 and D684, while in *S. typhimurium* an equivalent A:T pair is recognized by A684 and S687. We introduced K681A/D684S substitutions in *E. coli* BrxX. Similarly, *E. coli* BrxX with G785T/N823S substitution was constructed to mimic the recognition of the fourth

C:G pair in *S. typhimurium*. In the EOP assay, the K681D/D684T mutant retained the wild-type level of defense against the T7Δ0.3 phage but provided no defense against λ$_{vir}$. Unexpectedly, the G785T/N823S mutant provided enhanced defense against both phages (Fig. 4A). We used the Oxford Nanopore Technology (ONT) sequencing to assess the methylation specificity of the mutants (Fig. 4C). Contrary to expectations, both mutants methylated degenerate motifs: the BrxX$^{K681D/D684T}$ methylated BREX sites with any nucleotide except for the original G in the second position, while the BrxX$^{G785T/N823S}$ recognized sites with any nucleotide in the fourth position. To further confirm this observation, we performed a series of EMSAs with BrxX$^{WT}$ or BrxX$^{G785T/N823S}$ and DNA substrates carrying substitutions at the fourth position of BREX site (Fig. 4D). While WT BrxX demonstrated

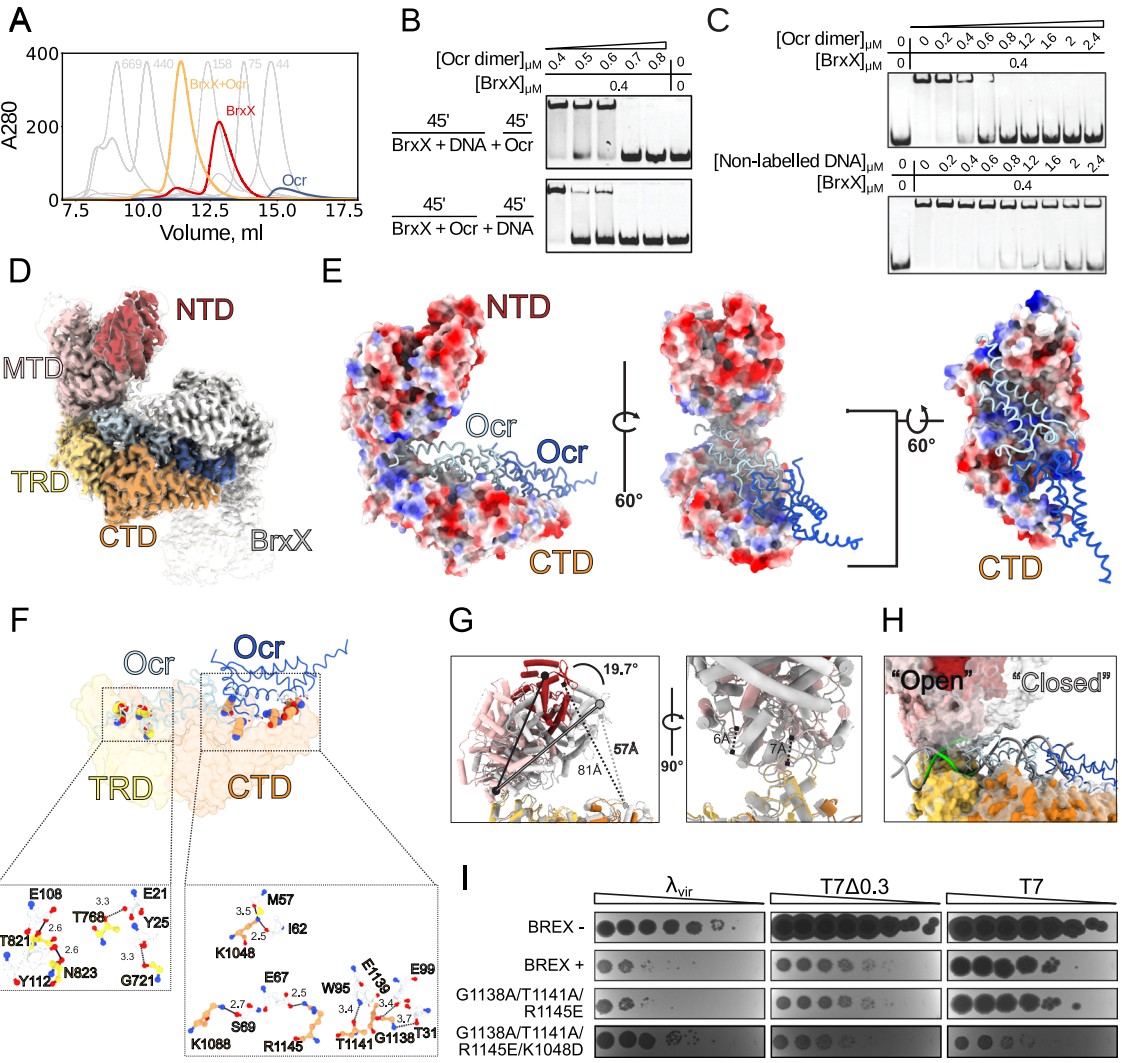

**Fig. 5 | The DNA mimic Ocr displaces DNA from BrxX and locks it in an inactive conformation. A** SEC traces of BrxX, Ocr, or an equimolar BrxX + Ocr mixture injected onto Superdex 200 Increase 10/300 column. SEC profiles of standard calibrants and their corresponding molecular masses are indicated in the background in gray. **B** EMSA with 20 nM Cy5-labeled 40 bp dsDNA substrate with a single BREX site incubated with a 20-fold (0.4 μM) molar excess of BrxX in the presence of indicated concentrations of Ocr dimer. BrxX was either incubated with DNA for 45 min, followed by an additional 45 min incubation with Ocr (top), or was first incubated with Ocr for 45 min, followed by an additional 45 min incubation with DNA (bottom). **C** Competition between DNA and Ocr for binding to BrxX. 20 nM Cy5-labeled 40 bp dsDNA substrate bearing one BREX site was incubated with a 20-fold (0.4 μM) molar excess of BrxX, followed by an indicated amount of Ocr dimer (top) or of a non-labeled dsDNA substrate (bottom). **D** CryoEM maps of the BrxX-Ocr complex. The autosharpened map, contoured at 7σ, is colored according to BrxX domains coloring scheme (Fig. 2) and shown within the transparent contour of the unsharpened map (at 5σ). Ocr monomers are shaded in blue; the lower-resolution BrxX monomer is shown in white. **E** Surface coloring based on Coulombic electrostatic potential (as implemented in ChimeraX), where red is negative potential and blue is positive potential. Ocr dimer is shown as representation. **F** Direct interactions between BrxX and Ocr residues. Interacting residues are shown as balls and sticks; Ocr residues are white while TRD and CTD residues are colored yellow and orange, respectively. **G** A structural superposition of atomic models of DNA-bound (white) and Ocr-bound (colored by domain) BrxX monomers. Movements are indicated. **H** Structural superposition of surface models. DNA-bound model is depicted in transparent white and an Ocr-bound depicted colored by domain. Alignment generated via ChimeraX matchmaker tool using the TRD domain boundaries. **I** EOP assay with WT T7, Ocr-deficient T7Δ0.3, and λ_vir phages and cells carrying BREX systems encoding indicated BrxX mutants.

strict specificity for the A at the fourth position, G785T/N823S variant was able to bind all four substrates, albeit with decreased affinity. Our results demonstrate that BREX site specificity could be engineered through structure-guided mutagenesis and could result in significant increase of anti-phage activity.

## Ocr sequesters two BrxX monomers and prevents DNA binding

DNA mimic Ocr encoded by T7 phage is an inhibitor and a trigger of various immune systems[41,58,59]. Ocr inhibits BREX defense by binding BrxX in vivo[40]. To understand the mechanism of this inhibition we studied interaction between BrxX and Ocr in vitro. We incubated BrxX with an excess of Ocr and analyzed the sample composition by size-

exclusion chromatography (SEC). A shift in the SEC peak of the complex compared to the BrxX monomer peak indicated formation of an oligomeric species with a molecular weight likely corresponding to a BrxX₂:Ocr₂ tetramer (Fig. 5A).

To demonstrate that Ocr competes with DNA for the binding to BrxX, we tested its ability to disrupt the BrxX:DNA complex formation using EMSAs. At our assay conditions, a 20-fold molar excess of BrxX was sufficient for the sequestration of all dsDNA in the reaction. Subsequent incubation with a 1.5-fold molar excess of Ocr dimer over BrxX completely abolished the BrxX:DNA complex formation and released free DNA (Fig. 5B). Thus, Ocr can efficiently displace DNA from preformed BrxX:DNA complexes. When BrxX was first incubated with Ocr

followed by the addition of DNA, an equimolar amount of Ocr dimer was sufficient to block the interaction of BrxX with DNA (Fig. 5B). To directly demonstrate that Ocr is a preferred substrate for BrxX, we compared the ability of Ocr or unlabeled dsDNA to compete with labeled dsDNA for the formation of the BrxX:DNA complex. We found that in both BLI and EMSA, Ocr was a better binding partner of BrxX (Fig. 5C, Supplementary Fig. S9A–C).

We next sought to determine the molecular basis of Ocr binding to BrxX by single-particle cryoEM analysis. 2D class averages (Supplementary Figs. S5F, S7 and S8) showed that the sample contained two distinct particle sets, enabling us to solve both the apo and Ocr-bound structures from a single dataset. 57% of the extracted particles yielded a 2.8 Å global resolution map (**BrxX:Ocr**) corresponding to a BrxX$_2$:Ocr$_2$ complex where two individual BrxX monomers interact with a dimer of Ocr via their TRDs and CTDs. The two BrxX monomers are oriented at an angle of 125° relative to each other; however, the complex displays imperfect symmetry due to the local flexibility of the individual components resulting in varied local resolution (Supplementary Fig. S7E). We resolved one monomer to ~2.8 Å (Fig. 5D) clearly defining all structural features except for a flexible loop between residues 395 and 420. The MTD displayed density for a molecule of SAM, which we modeled based on the higher resolution DNA-bound structure. The map resolution for the second BrxX monomer stretched from ~3.4 Å around the core of the particle to ~4–6 Å at the periphery (Supplementary Fig. S7E); hence, we rigid-body fitted the first monomer model into the density of the second monomer (see "Methods"). Around ~10% of BrxX particles did not bind Ocr in our sample, enabling us to determine the structure of apo BrxX by rigid-body fitting BrxX:DNA model in a 3.9 Å BrxX$_{apo}$ map (Supplementary Fig. S8 and "Methods"). Given that apo BrxX adopts exact same structural conformation as BrxX bound to Ocr (Supplementary Fig. S8A, B), we proceeded with the higher resolution structure for all further analyses.

Ocr is highly negatively charged, and its interaction with BrxX is largely driven by electrostatic contacts with the positively charged groove on the lower half of the BrxX monomer (Fig. 5E). To form direct contacts, Ocr utilizes two pairs of tyrosine and glutamic acid residues that form hydrogen bonds with four residues of BrxX TRD: T821, N823, T768, and the main-chain carbonyl of G721 (Fig. 5F). Three out of four of these TRD residues are also directly involved in DNA recognition or binding: N823 directly interacts with the A4 in the target strand of the recognition site, while the main-chain amide of G721 interacts with the phosphate backbone of T5′ base in the non-target strand. T768 lies between the two residues interacting with a terminal phosphate of the non-target strand, S767 and T769 (Fig. 3A). Ocr is additionally stabilized by the BrxX CTD via a salt bridge between Ocr E67 and BrxX R1145; the latter residue also interacts with the DNA away from the recognition site. Other CTD interactions include hydrogen bonding between a number of residues (Fig. 5F). We mutated the Ocr-binding interface of BrxX CTD and tested the mutants in EOP against the Ocr-deficient T7 *Δ0.3* and WT T7 phage (Fig. 5I). The BrxX$^{G1138A/T1141A/R1145E/K1048D}$ provided equivalent defense against WT T7 phage as BrxX$^{WT}$ to the Ocr-deficient phage showing that the Ocr inhibition was relieved. Triple mutant BrxX$^{G1138A/T1141A/R1145E}$ did not show resistance to T7, highlighting the critical role of K1048 in Ocr binding. This positively charged residue is however also involved in non-specific DNA recognition, and the same T7-resistant mutant demonstrated reduced protection against $\lambda_{vir}$.

Structural alignment of the TRD domain in BrxX-DNA and BrxX-Ocr complexes results in Ocr and DNA occupying the same horizontal plane, allowing for a comprehensive comparison of DNA and Ocr geometry and key residue interactions (Fig. 5F). Structural alignments of each of the four BrxX domains between the two conformations show that the NTD rearranges the least, and the CTD rearranges the most, however all RMSD values fall below 2 Å (Supplementary Table S5). The superposition of the two structures makes it clear that it

is the "upper jaw" of the BrxX monomer that closes around the DNA upon recognition as the MTD moves 19.7° between the open and closed states (Fig. 5G, left). The MTD loops that stabilize distorted DNA move 6–7 Å from the open to the closed state (Fig. 5G, right). The resolved 25 bp DNA fragment has the same width as the Ocr monomer (Fig. 5H); however, compared to the relatively rigid Ocr, DNA is a more flexible substrate that can bend and deform. While DNA is clearly able to associate with BrxX in the open conformation, stabilization of a flipped adenine is only possible in the closed conformation which brings each strand of the DNA substrate in close contact with BrxX, and facilitates the formation of stabilizing contacts between the MTD and the distorted phosphate backbone of bound DNA.

## BrxX alone is not an active methyltransferase in vitro

S-adenosyl-L-methionine (SAM) is a universal methyl group donor to C, O, N, or S atoms and thus an essential cofactor of methyltransferases[60]. In both DNA-bound and Ocr-bound BrxX structures, a clear density for a SAM molecule is visible deeply nestled within the hydrophobic pocket in the core of the MTD (Fig. 6A). Apart from the narrow channel leading from the flipped base pocket to the active site, the SAM molecule is otherwise completely surface inaccessible. The lack of significant rearrangement between the apo- and DNA-bound conformations demonstrates that BrxX, at least in the presence of SAM, is in a methylation-primed state, and has a binding pocket pre-organized to receive the flipped base. This observation is in line with the EMSA results showing the enhancing effect of SAM on DNA binding (Fig. 1C), and with our cryoEM data. SAM was critical to observe high-resolution 2D classes, suggesting that SAM binding limits flexibility and helps to organize the MTD. All three regions of the SAM molecule (adenine ring, ribose, methionine moiety) make extensive contacts with BrxX (Fig. 6B). The adenine ring is stacked between F540 and I355 and further stabilized by van der Waals contacts with L536 and I239. The ribose ring forms direct hydrogen bonds with Q244, and the methionine end of the molecule is also encapsulated within an extensive network of H-bonding (Fig. 6B). The NPPY motif IV sequence (508–511) accommodates the target adenine flipped out of the DNA helix, with Y511 playing a key role in adenine stabilization. In line with these observations, the Y511A substitution disrupted both site-specific and non-specific DNA binding (Fig. 6C), and completely abolished the BREX defense and methylation in vivo, without causing cell toxicity (Fig. 6D). Inflection temperature analysis using Tycho NT.6 confirmed that the Y511A mutation did not affect the overall stability of the protein (Supplementary Fig. S9D). These results indicate that anchoring of the MTase domain by a flipped-out adenine following the transition of BrxX to the closed conformation is an essential step in BREX site recognition.

To detect BrxX methyltransferase activity in vitro we relied on the fact that the AluI restriction endonuclease is sensitive to adenine methylation[61], and that the AluI recognition site (**AG**C**T**) overlaps with the BREX site (**GGTA**A**G**). To demonstrate AluI sensitivity to BREX methylation we constructed a plasmid carrying an overlapping BREX/AluI site and confirmed that when purified from BREX$^+$ *E. coli* HS cells, this site is not cleaved by AluI in contrast to the same plasmid purified from BREX$^−$ cells (Fig. 6E). Next, we used a 40 bp dsDNA substrate carrying a single BREX/AluI and a single Dam site in the in vitro methylation reaction with wild-type BrxX or BrxX$^{Y511A}$ mutant. Dam methyltransferase was used as a positive control. While Dam methylation was complete after overnight incubation with the protein, as revealed by the sensitivity of methylated DNA to DpnI/II cleavage (Supplementary Fig. S9E), substrate incubated with BrxX was efficiently cleaved by AluI, indicating the lack of methylation (Fig. 6F). To exclude the possibility that in our conditions BrxX methylates its target sites at a slow rate that is beyond the detection limit of a restriction-sensitivity assay, we attempted to detect N6-methyl-adenine (m6dA) using HPLC-MS analysis of DNA substrate digested to nucleosides

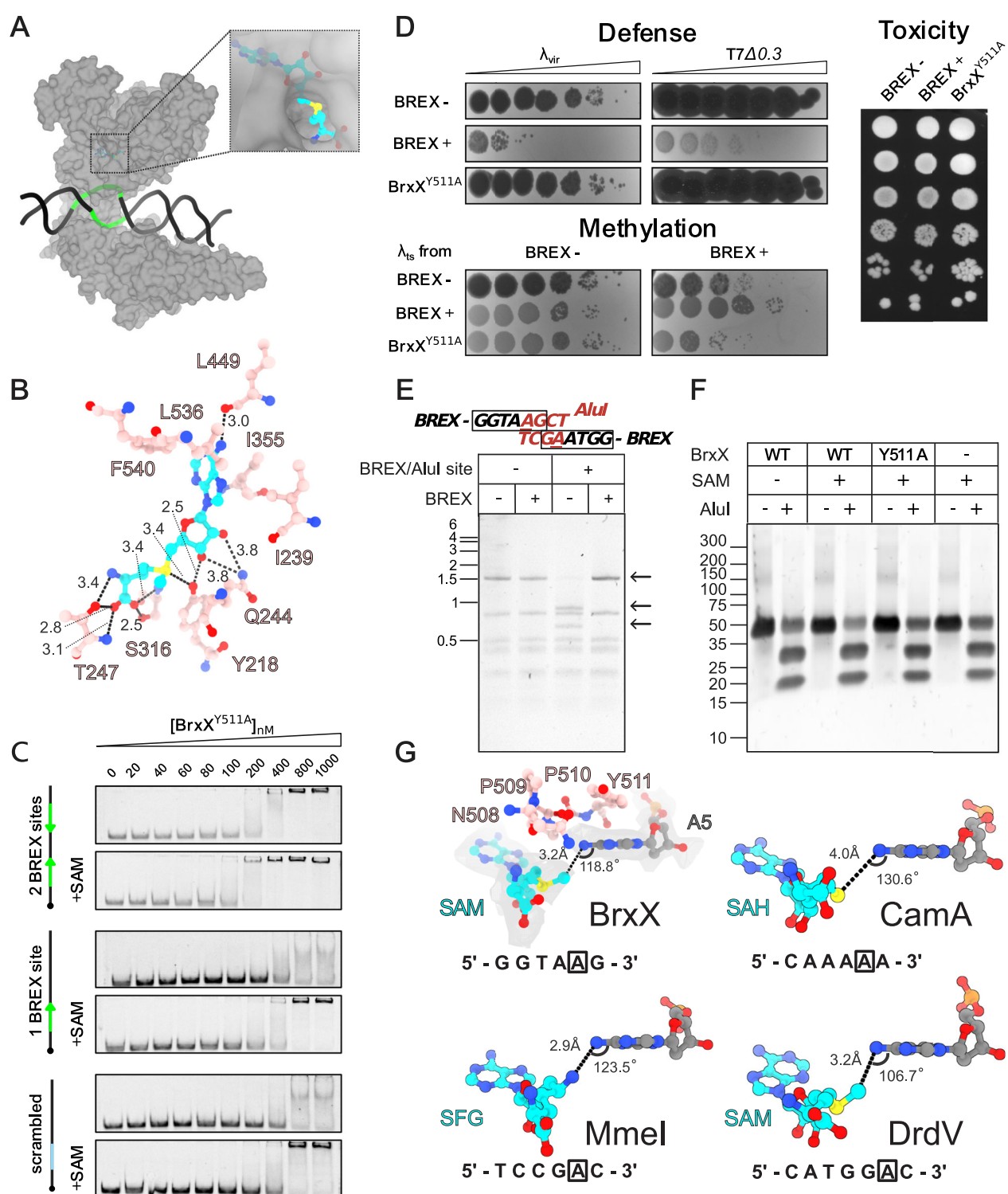

(Supplementary Fig. S9F). Again, while the amplitude of the m6dA signal in the Dam-treated sample corresponded to full methylation of a single Dam site of the substrate, no signal was detected in the BrxX-treated sample, confirming that BrxX alone lacks methyltransferase activity at the conditions of our in vitro assay.

In the DNA-bound BrxX structure, the N6 amino group of the extrahelical adenine forms hydrogen bonds with the side chain of N508 (4.3 Å) and the backbone carbonyl of T243 (3.6 Å). This would negatively polarize the N6 atom and activate it for a direct transfer of the methyl group from SAM via nucleophilic substitution using Sn2

proximity and desolvation mechanism. However, for this to occur, the three atoms (N, S, C) must form a linear arrangement. As the two catalytic moieties are at an angle of 118.8° relative to one another, methylation in this state would not be possible indicating a pre-, or post-methylation state. This is no different from other similar methyltransferases such as MmeI[54], CamA[55], and DrdV[29], where the structures show a similar distance and angle orientation of the N6-adenine atom and the methyl group carbon atom (or the equivalent moiety in the respective structures) (Fig. 6G). None of these structures directly show how the methylation reaction occurs, but as all these

**Fig. 6 | BrxX alone lacks methyltransferase activity. A** Surface representation of the BrxX monomer shown in gray bound to DNA (black) containing the BREX binding site (green). Inset highlights the only external access point to the SAM molecule enclosed within the catalytic pocket (cyan backbone, colored by heteroatom). **B** Direct interactions between the SAM molecule and MTD residues. Dashed lines indicate hydrogen bonding with distances indicated in Angstroms. **C** EMSA with 20 nM Cy5-labeled 43 bp dsDNA substrate without BREX sites ("scrambled"), or with one or two BREX sites, incubated with the indicated amount of BrxX[Y511A] without co-factors or in the presence of 0.5 mM SAM. Green arrows indicate the orientation of BREX sites. Compare with Fig. 1C for EMSA with wild-type BrxX performed in identical conditions. **D** Effects of BrxX[Y511A] mutation on BREX defense, methylation, and toxicity. BREX defense is demonstrated by an EOP assay with BREX-sensitive $\lambda_{vir}$ and T7$\Delta 0.3$ phages. BREX methylation is estimated by the ability of $\lambda_{ts}$ induced from the indicated lysogenic cultures to plaque on BREX⁻ and BREX⁺ lawns (see "Methods"). Toxicity was estimated in a drop-spot test on LB agar plates. BrxX[Y511A] was introduced in the context of the full BREX cluster. All assays were performed in biological triplicates and representative plates are shown. **E** Overlapping BREX methylation inhibits AluI cleavage. An agarose gel shows AluI digest of pHERD30t or pHERD30t bearing an overlapping BREX/AluI cleavage site purified from either BREX⁺ (pBREX AL) or BREX⁻ (pBTB) cultures. Arrows indicate the fragment bearing overlapping BREX/AluI site and its AluI cleavage products. **F** AluI cleavage with 40 bp dsDNA substrate containing an overlapping BREX/AluI site and incubated with 100 µM BrxX overnight. **G** A comparison of orientations of the flipped adenine and SAM or an equivalent ligand between BrxX (this work), CamA (PDB: 7LT5), MmeI (PDB: 5HR4), and DrdV (PDB: 7LO5). The 5′−3′ recognition site with the methylated adenine boxed is indicated for each protein. In the BrxX structure, the sharpened map density contoured at 5σ is shown surrounding each molecule, with the catalytic residues (NPPY motif) from the MTD shown in pink.

enzymes methylate their substrates in vitro, we conclude that the elements present only in BrxX (such as NTD, CTD, or the MTD insert) might act as cis-inhibitory elements preventing BrxX from occupying a transient methylation-competent conformation and that interactions with external factors are required to relieve this inhibition.

## BREX methylation and defense require the assembly of a macromolecular BrxBCXZ complex

To identify conditions supporting BREX methylation in vivo, we expressed BrxX (alone, or in combination with other BREX proteins) in *E. coli* cells lysogenized with the $\lambda$ phage. Induction of a prophage from BREX⁺ cells resulted in the production of methylated phage progeny that bypassed BREX, while the non-methylated phage produced by the BREX⁻ culture was restricted on the BREX⁺ lawn[18] (Fig. 7A). Phage induced from cells expressing BrxX alone was fully susceptible to BREX defense, indicating the lack of BREX methylation (Fig. 7A) in line with in vitro results (Fig. 6E, F). Co-overproduction of BrxX with any other single BREX protein also did not support in vivo methylation. However, phages induced from cells simultaneously producing BrxB, BrxC, BrxX, and BrxZ proteins were resistant to BREX (Fig. 7A), consistent with our previous observation that individual deletion of each of these components abolished BREX methylation[18]. This result shows that the methylation-competent state of BrxX requires BrxB, BrxC, and BrxZ proteins, and suggests that the four proteins may physically interact.

To systematically determine protein–protein interactions of BREX components we introduced a sequence encoding Strep-tag II onto the 3′ end of each *brx* gene individually within the context of a full BREX gene cluster. Addition of Strep-tags to BrxA and BrxC disrupted phage defense (Supplementary Fig. S9G). We therefore carried pull-down assays only with C-terminally Strep-tagged BrxB, BrxX, BrxZ, and BrxL constructs (Fig. 7B). The results support the existence of a BrxBCXZ complex detected when BrxZ-Strep is used as a purification bait. SEC analysis of BrxZ-Strep eluate showed two separate peaks corresponding to BrxZB and BrxBCXZ complexes (Fig. 7C, D). The BrxBCXZ complex co-purified with cellular DNA and thus migrated close to the void volume of a gel-filtration column, preventing the determination of molecular weight and stoichiometry of the complex. Pull-down with BrxB confirmed that this protein directly binds to BrxZ, resulting in the formation of a stable ~200 kDa complex (Fig. 7C, D). BrxX-Strep co-purified with BrxC, and also with the host DNA, however, DNase I treatment allowed us to estimate the molecular weight of BrxCX as ~760 kDa (Fig. 7C–F). While the exact determination of the size of BrxCX and BrxBCXZ complexes is not possible due to the limited resolution of SEC, the results indicate the formation of oligomers involving BrxC. Compared to other tagged BREX proteins, the level of BrxL-Strep production was the lowest and only purification from a larger volume of culture allowed us to detect its interaction with BrxX (Fig. 7C, D).

Together, these results support a structural model for the assembly of a BREX complex where BrxB preferentially interacts with BrxZ, and BrxX preferentially interacts with BrxC. Given that all four proteins are required for both BREX defense and methylation[18], a single BrxBCXZ complex likely combines both activities and relies on BrxX for recognition of non-methylated BREX sites in DNA, resulting in either methylation of host DNA, or exclusion of invading mobile elements. We hypothesize that conserved NTDs and CTDs of BrxX are required for the interactions with BrxZ and BrxC proteins and that these interactions alter the conformation of BrxX to make it methylation-competent. How the BREX system coordinates defense and methylation activities within a single complex and whether it experiences structural re-arrangements in the process of invading DNA exclusion represents an important question for further studies.

## Discussion

BREX is a representative of a cohort of immunity systems that encode DNA modification components yet lack a known endonuclease that could destroy unmodified DNA. Type I BREX systems epigenetically modify specific sequences (BREX sites) in host DNA to discriminate it from unmodified DNA of genetic invaders. We show that BrxX is the only BREX component that can recognize BREX sites in vitro. BrxX also binds to non-methylated BREX sites in phage DNA in vivo. Thus, BrxX must be involved in foreign DNA sensing and the initiation of BREX defense, as well as in the establishment of protective methylation of host DNA. The dual role of BrxX as a host DNA methylase and a phage infection sensor is confirmed by an observation that a mutation in the methylase active site (Y511A) inactivates both methylation and BREX defense but does not result in toxicity, as would be expected for a system with independent modification and restriction modules. In addition, mutations that prevent DNA binding by BrxX abolished BREX defense, while a mutation that relaxes the specificity of BREX site recognition by BrxX increased the level of defense (Fig. 4A). These results align well with the recently reported scanning mutagenesis of *Salmonella enterica* BrxX (PglX), where multiple non-toxic BrxX variants were obtained that retained the in vivo methylation activity but compromised defense[34]. Together, these results link BREX defense with site-specific DNA recognition by BrxX; the mechanism controlling whether BrxX methylates DNA (in case of host DNA following replication) or triggers subsequent BREX defense that limits the propagation of invaders with unmodified DNA is yet to be determined, as is the mechanism of BREX defense.

We have determined the apo, Ocr- and DNA-bound structures of *E. coli* BrxX. We show that BrxX contains MTD and TRD homologous to those of Type IIL R-M enzymes, and, in addition, contains NTD, CTD, and insertion loop in the MTD specific to BrxX proteins. Structural alignment of BrxX in different conformations provides additional insights into events preceding the methylation reaction (Fig. 8). The TRD and MTDs of BrxX are connected by a flexible linker, and the apo-protein exists in an open conformation. BrxX binds DNA

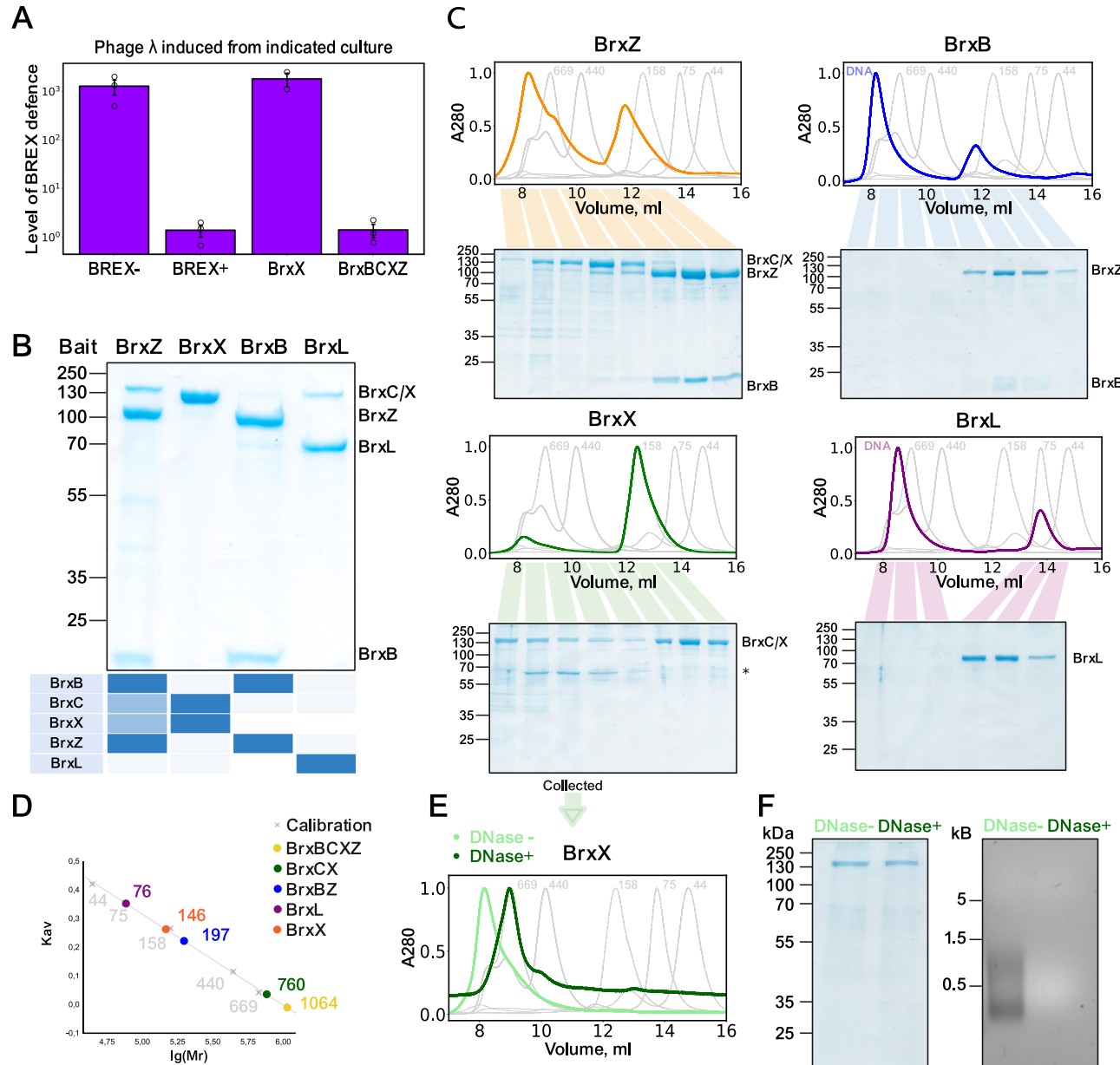

**Fig. 7 | BREX methylation requires the assembly of a macromolecular BrxBCXZ complex. A** In vivo BREX methylation requires co-production of BrxX, BrxC, BrxZ, and BrxB. Prophage $\lambda_{wt}$ was induced from indicated lysogens and the status of its BREX methylation was determined by plaquing on BREX⁺ and BREX⁻ lawns. The level of BREX defense calculated from the EOP performed in biological triplicates with independently induced $\lambda_{wt}$ phages. Data are presented as mean values ± SEM. **B** In vivo pull-downs with Strep-tagged BREX proteins expressed in the context of the full BREX cluster. Strep-tagged proteins (baits) are indicated at the top and co-eluted proteins, identified through MALDI-TOF mass spectrometry, are shown on the heatmap below. The Strep-trap column eluates were concentrated, and proteins resolved by 4–20% gradient SDS-PAGE. **C** Strep-tag pull-down eluates (**B**) were concentrated and separated on a Superdex 200 Increase 10/300 column. For each run, 280 nm absorbance was normalized to the highest value. SEC profiles of calibrants and their corresponding molecular masses are indicated on the background in gray. An asterisk indicates contaminant protein. DNA non-specifically binds the StrepTrap HP column, creating peaks eluting with the void volume, which are the most evident in BrxL and BrxB runs. **D** SEC calibration curve with estimated masses of obtained protein complexes. **E** DNAse I treatment of fractions containing the BrxCX complex results in the reduction of apparent complex size. **F** SDS-PAGE results (left) demonstrating protein and agarose gel electrophoreses (right) demonstrating DNA content of the BrxCX peak before and after DNase I treatment. In all panels, the protein or DNA molecular weight markers are shown on the left.

non-specifically through electrostatic interactions of the DNA phosphodiester backbone with the TRD and CTD (**Stage I**). The DNA mimic protein Ocr exploits this non-specific DNA recognition property to sequester BrxX (**Stage VI**). Thus, Ocr interaction with BrxX homologs does not depend on a system-specific and highly variable recognition site, explaining its broad inhibitory activity. Following non-specific DNA binding, BrxX might search for the adjacent BREX site and recognize it through direct sensing of bases in the DNA major groove followed by the establishment of an extensive network of additional TRD-DNA contacts (**Stage II**). The flipping of the fifth adenine upon the closure of the flexible MTD is concomitant with the recognition of the last, sixth base by the closed MTD resulting in a stable pre-methylation complex (**Stage III**). Substitution of BrxX Y511, which forms a $\pi$-stacking contact with the extrahelical adenine, results in a loss of both DNA binding in vitro and BREX defense in vivo. Thus, the pre-methylation complex is required for stabilization of the BrxX:DNA interaction and productive BREX site recognition.

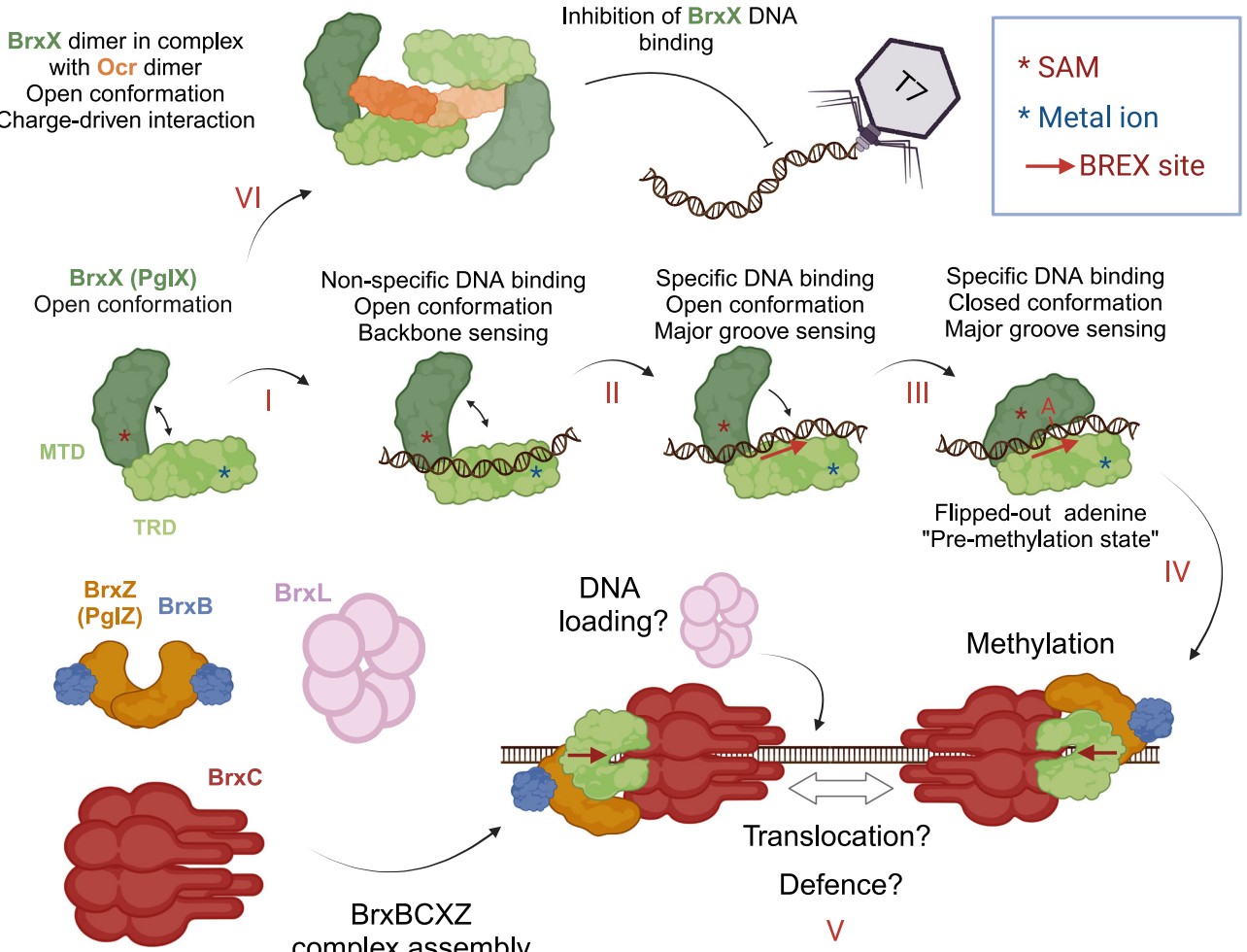

**Fig. 8 | A model of foreign DNA sensing by BrxX and its inhibition by Ocr.** BREX sites recognition requires target adenine flipping and BrxX conformation transition from an open to the closed state (**I, II, III**). Ocr locks two BrxX monomers in an open conformation unable to bind DNA (**VI**). BREX DNA methylation (**IV, V**) requires the assembly of the BrxBCXZ complex, which might be also responsible for BREX defense. Stages **I–VI** are also discussed in the text. This figure was created with BioRender.com.

Stabilization of the BrxX:DNA complex also explains the role of SAM in BREX defense. In Type I R-M systems, SAM serves not only as a donor of methyl groups but also as a cofactor of the restriction reaction[42]. We show that the BrxX:DNA interaction is SAM-dependent: SAM moderately enhances non-specific DNA binding but has a stronger effect on BrxX binding to BREX sites. This could be explained by an overall stabilization of the BrxX methyltrasferase domain in the presence of SAM, and an improved ability to accommodate flipped-out adenine in the catalytic pocket. Since BrxX is involved in sensing non-methylated BREX sites in phage DNA, SAM depletion should decrease the ability of BrxX to recognize foreign DNA, rendering BREX defense inefficient. This explains our previous observation of BREX defense inhibition by phage T3 SAM lyase[43] and highlights the role of homologous viral enzymes as efficient inhibitors of SAM-dependent immunity systems.

We were unable to demonstrate methyltransferase activity of BrxX alone either in vitro or in vivo. Co-expression of BrxX with BrxB, BrxC, and BrxZ is required for BREX methylation in vivo. The organization of the BrxX catalytic pocket in the DNA-bound structure is similar to those observed in the structures of stand-alone methyltransferases bound with SAM, SAH, or sinefungin[29,54,55] (Fig. 6G). We suggest that the structures of such active enzymes likely capture either a pre-methylation or a structurally similar post-methylation conformation. We speculate that BrxX CTD or NTD domains inhibit the

transition to a methylation-competent state, and only upon the interaction with other BREX proteins, the conformation changes to license methylation of target DNA (**Stage IV**). Some evidence to support this notion comes from the study of PglX from the Pgl phage immunity system of *Streptomyces coelicolor* A3(2) that lacks NTD and CTD homologous to BrxX and actively methylates DNA in vitro[23].

We show that BrxB, BrxC, BrxX, and BrxZ are jointly required for methylation and BREX defense. We also show that these proteins form a complex (**Stage V**). Combining both self/non-self recognition and defense functions in one complex is highly advantageous for an immune system that methylates only one DNA strand within an asymmetric recognition motif. Since non-methylated BREX sites emerge in host DNA after each round of replication, BREX defense must be tightly regulated to avoid potentially lethal activation at host non-methylated sites and discriminate them from non-methylated sites in the phage DNA. Newly synthesized DNA of the host will retain methylation on the template strand; thus, if to consider two neighboring BREX sites located on opposite DNA strands, one site will always be methylated, while the phage DNA is completely devoid of BREX methylation on both strands. Similar to other immune systems that methylate only one DNA strand[24,28], discrimination between self/non-self DNA might require simultaneous binding of BrxBCXZ complexes to BREX sites located on the opposite DNA

strands, followed by interaction between the two complexes. The BrxC ATPase is homologous to ORC/Cdc6 (Origin Recognition Complex) proteins involved in the initiation of archaeal and eukaryotic DNA replication[36,37] and may play a role of the assembling scaffold/recruitment unit, similar to that of ORC, which loads the replicative helicase MCM onto replication origins[62,63]. Strikingly, the BrxL protein is structurally similar to MCM helicases[38] and thus could represent a BREX translocase. BREX systems that lack BrxL encode BrxH helicase instead[17]. BrxL is not required for DNA methylation and defense in *S. typhimurium* Type I BREX[57] but was shown to be required for BREX defense in *Acinetobacter* 394 system[38]. Another subunit of the BREX complex, BrxZ (PglZ), is homologous to the effector of the PorXY two-component system[35]. The PglZ domain of PorX is a phosphodiesterase cleaving cyclic and linear oligonucleotides[35] and BrxZ may be responsible for the effector response to phage infection.

To summarize, in this work we have described the molecular basis of site-specific DNA recognition by BREX anti-phage defense system, the first step in BREX-mediated immune response. Our results show that in addition to the methyltransferase function, BrxX is strictly required to sense phage infection. We have also revealed how anti-defense protein Ocr sequesters and inactivates BrxX. Finally, we have demonstrated that both BREX defense and methylation require a supramolecular BrxBCXZ complex. In contrast to Type II R-M systems, which are evolutionary constrained by the necessity for mutations to accumulate simultaneously in both the R and M components, novel BREX specificities could emerge upon a single mutation in in the BrxX TRD and result in significantly enhanced antiviral defense (Fig. 4). This implies that BREX could more efficiently counteract accumulation of escaper phages through changes of its recognition motif and might explain previously observed instances of duplications or phase variation of *pglX* gene in BREX clusters[17]. Our DNA-bound structure provides the molecular basis for BREX DNA recognition and can be used as a blueprint for the design of BREX variants with desired target specificities.

## Methods

### Bacterial strains and bacteriophages
All bacterial strains, plasmids, and bacteriophages used in this study are listed in Supplementary Table S2. The majority of experiments were performed with *E. coli* BW25113. BREX⁻ is a BW25113 carrying an empty pBTB-2 vector, while BREX⁺ carries a full 6-gene *E. coli* HS BREX cluster cloned in pBTB-2 under native promoters[18]. Unless stated otherwise, bacterial cultures were propagated in LB medium (Lysogeny Broth: 10 g/L NaCl, 10 g/L tryptone, 5 g/L yeast extract) at 37 °C with appropriate antibiotics. $\lambda_{vir}$ is an obligatory lytic mutant of the phage $\lambda$. $\lambda_{ts}$ is a *cI857 bor::Cm* variant encoding thermosensitive CI repressor, allowing for a temperature-controlled lysogen induction. For the infection with phage $\lambda$ cells were supplemented with 0.2% maltose and 5 mM MgSO₄. T7Δ*0.3* lacks *0.3* gene encoding anti-BREX protein Ocr and thus is sensitive to BREX defense. Phages T4 and T4₁₄₇ were propagated on *E. coli* DH10B *galU⁻* cells and produced for genomic DNA purification in liquid culture. $\lambda_{ts}$ for genomic DNA purification was induced from BREX⁻ and BREX⁺ BW25113 lysogens. Cells were grown to OD₆₀₀ = 0.8 in 100 mL LB media at 30 °C, followed by 1:1 mixing with 1.2% cooled-down LB agar, 15 mL of the mixture was poured onto Petri dishes, and set for the phage production at 42 °C overnight. The next day phage-containing agar was scrapped out from the plates into 50 mL falcon tubes and centrifuged 3 times at 4 °C for 10 min at 12,000 × *g*. Supernatants from each centrifugation were collected, and treated with 50 μL chloroform/10 mL of lysates.

### Phage DNA purification
High-titer phage lysates (>10⁹ pfu/mL) of $\lambda_{ts}$ and T4 phages were used for genomic DNA purification. Phage particles were precipitated with

10% polyethylene glycol 8000, 1 M NaCl overnight at 4 °C with shaking, and DNA was extracted with phenol:chloroform according to a standard procedure[64].

### Plasmid construction
Primers used to generate plasmids are listed in Supplementary Table S3. pBAD vectors for His-tagged BREX proteins expression and Strep-tag variants of the pBREX AL vector were obtained previously[18,40]. Y511A mutation in pBAD-His₆-BrxX was introduced using Q5 site-directed mutagenesis kit (NEB), while all other mutations in the pBAD-His₆-BrxX or pBREX AL were introduced by joining two plasmid fragments, carrying mutation in the overlap region, using NEBuilder HiFi DNA assembly kit (NEB). pBAD-His₆-Ocr was obtained by cloning *0.3* gene from T7 phage into pBAD-HisB[40] vector using Gibson Assembly Master Mix (NEB). Introduction of the overlapping BREX/AluI site to the pHERD30t vector was also achieved via Q5 site-directed mutagenesis kit (NEB). Plasmids were transformed into chemically competent XL1-Blue cells (Evrogen). The presence of desired mutations was verified by Sanger sequencing with specific primers, while the absence of secondary mutations was confirmed via whole-plasmid BGI sequencing.

### Monitoring dynamics of phage DNA injection with a potassium efflux assay
Monitoring potassium efflux requires culture infection at high MOI. First, we identified that at MOI of 5 potassium efflux can be robustly detected, while BREX anti-phage defense is still efficient. $\lambda$ $cI_{857}$ $S_{am7}$ was used to produce high-titer stock (~2 × 10¹² pfu/mL). A $\lambda$ $cI_{857}$ $S_{am7}$ lysogen was grown in 30 mL LB at 30 °C for 2 h 30 min, and the prophage was inducted with a 5 min heat shock at 45 °C in a water bath. Cells were then incubated at 37 °C for 2 h for phage particle production. $\lambda$ $cI_{857}$ $S_{am7}$ cannot lyse the membrane, allowing phage particles accumulation inside the cell. Cells were collected by centrifugation (6000 × *g*, 10 min, 4 °C), resuspended in 2 mL of SM3 buffer (10 mM Tris-HCl pH 7.5, 10 mM NaCl, 4 mM MgSO₄), and lysed by sonication. The cell debris was then removed by centrifuging the lysate for 10 min at 6000 × *g*. The resulting phage lysate was dialyzed overnight at 4 °C against 1 L of SM3 buffer to remove residual K⁺ and the titer was measured at 37 °C on a lawn of the *E. coli* amber suppressor strain QD5003.

For potassium efflux assays, BW25113 cells were grown at 37 °C in 15 mL LB supplemented with 0.2% maltose and 10 mM MgSO₄ to an OD₆₀₀ of 0.5. A 5 mL aliquot was removed, the cells were collected by centrifugation at 3000 × *g* for 5 min, and the resulting cell pellet was resuspended in 5 mL of SM3. The centrifugation/resuspension step was repeated three times to completely remove residual K⁺ ions.

The Orion potassium selective electrode (Thermo Scientific) was assembled and stored in 10 mM K⁺ buffer for 24 h. Before the assay, the electrode was adopted to low concentrations of K⁺ (10⁻⁷–10⁻⁵ M K⁺ in buffer with Ionic Strength Adjuster). A standard curve of K⁺ ions concentration was built before each experiment in a range of 10⁻⁶–10⁻³ M of K⁺. The electrode was equilibrated with 2 mL of cells for 5 min and then 100 μL of dialyzed phage was added, followed by brief mixing of the measuring cell. The signal was monitored for 2 h and extracellular K⁺ concentration was calculated using the standard curve. A BW25113 Δ*lamB* mutant strain that lacks the receptor for $\lambda$ adsorption was used as a negative control, and the experiment was performed in biological triplicates.

### Purification of recombinant proteins
N-terminally hexa-His-tagged BREX or Ocr proteins were expressed in BW25113 cells transformed with pBAD-His₆-BrxN/pBAD-His₆-Ocr (where N indicates one of the six BREX proteins) or the corresponding pBAD-His₆-BrxXᴺ derivatives (where N indicates mutations listed in the Supplementary Table S1). Since the

presence of N-His in BrxX, and N-His-TEV in Ocr did not interfere with the in vivo or in vitro functions of the protein, a non-processed variants were used throughout the work. Overnight culture obtained from the freshly transformed cells was diluted 100× into LB media (8 L) supplemented with ampicillin, grown at 37 °C until $OD_{600}$ reached 0.9, and induced with L-arabinose (0.2% w/v final concentration). Following overnight expression at 18 °C, bacteria were harvested by centrifugation (10 000 g, 4 °C), and the pellet was resuspended in a buffer A (50 mM Tris-HCl pH 8.0, 300 mM NaCl, 20 mM imidazole, 5 mM 2-mercaptoethanol, 5% glycerol) supplemented with a cOmplete Protease Inhibitor Cocktail (Roche). Cells were lysed by sonication and the lysate was centrifuged (21,000 × $g$, 4 °C). The clarified lysate was applied to 5 mL HisTrap HP columns (Cytiva) using an ÄKTA pure FPLC system (Cytiva). The resin-bound protein was first washed with 8 column volumes (CV) of buffer A, followed by gradient elution with buffer B (50 mM Tris-HCl pH 8.0, 500 mM NaCl, 500 mM imidazole, 5 mM 2-mercaptoethanol, 5% glycerol). BrxB and BrxZ were dialyzed overnight at 4 °C against GF buffer (50 mM Tris-HCl pH 8.0, 150 mM NaCl, 5 mM 2-mercaptoethanol) and concentrated for gel-filtration (see below) directly after this step, whilst BrxA, BrxC, BrxX, BrxX$^{Y511A}$, BrxL and Ocr were first purified on a heparin affinity column. For this, fractions corresponding to the protein peak were pooled and carefully diluted with buffer C (50 mM Tris-HCl pH 8.0, 5 mM 2-mercaptoethanol, 5% glycerol) until the concentration of NaCl reached 100 mM. The sample was loaded onto a 5 mL HiTrap Heparin column (Cytiva) and washed with five CVs of buffer C supplemented with 100 mM NaCl. Proteins were eluted by a gradient of buffer D (50 mM Tris-HCl pH 8.0, 5 mM 2-mercaptoethanol, 1 M NaCl, 5% glycerol). Ocr protein was collected from the flowthrough fractions, while other proteins were bound to the column. Therefore, Ocr was additionally purified via a second pass through a HisTrap HP column. Protein-containing fractions were collected and concentrated/buffer exchanged to GF buffer using 10-, 30- or 100-kDa Amicon centrifugal filter units (Merk), depending on the protein size. Concentrated proteins were loaded onto HiPrep 26/600 Superdex 200 size-exclusion column (Cytiva) connected to an ÄKTA pure FPLC system and the run was carried in a GF buffer. Protein-containing fractions were analyzed by SDS-PAGE, pooled, concentrated, distributed into 5 μL aliquots and snap-frozen in liquid nitrogen for storage at −80 °C for further in vitro studies. The protein concentration was measured using 280 nm absorbance with NanoDrop 8000 (Thermo Scientific).

## Electrophoretic mobility shift assay (EMSA)
Oligos used in EMSA are listed in Supplementary Table S3. To anneal dsDNA duplexes, equal amounts of 100 μM 5′-Cy5-labeled forward strand and unlabeled reverse strand oligonucleotides (Evrogen/Merck) were mixed in the annealing buffer (10 mM Tris-HCl pH 8.0, 50 mM NaCl). After heating to 95 °C for 10 min, the mixture was allowed to slowly cool to room temperature, before dilution in deionized water to achieve 100 nM final stock concentration.

**BREX proteins binding to non-specific and specific DNA substrates.** Proteins (in a range of concentrations), supplemented with 0.5 mM SAM or SAH, if necessary, were mixed with 20 nM Cy5-labeled DNA in EMSA buffer (0.5× TBE pH 8.0, 35 mM NaCl, 5% glycerol) in final 12 μL reactions and incubated for 30 min at room temperature. A substrate with a methylated BREX site was ordered through Sigma.

**EMSA of BrxX and DNA in the presence of Ocr.** The procedure was the same as above, 750 nM BrxX was incubated with either 20 nM DNA (bearing one BREX site) or Ocr for 45 min at room temperature, followed by the addition of either Ocr or 20 nM dsDNA and incubation for

further 45 min at room temperature. Ocr molar concentration was calculated for the Ocr dimer.

**Competitive binding of Ocr and DNA to BrxX.** Four hundred nanomoles BrxX were incubated with 20 nM 43 bp Cy5-dsDNA (bearing one BREX site) for 20 min at room temperature, followed by addition of either Ocr or unlabeled 43 bp dsDNA (bearing one BREX site) and incubation for 20 min at room temperature. Ocr molar concentration was calculated for the dimer.

Binding reactions for experiments listed above were run on 10% polyacrylamide (37.5:1) TBE gels for 35 min at 110 V at room temperature. Cy5-labeled DNA was visualized with the Molecular Imager Gel Doc XR System (BioRad).

**BrxX binding to phage genomic DNA.** BrxX (in a range of concentrations), supplemented with 0.5 mM SAM was mixed with either 130 pM $\lambda_{ts}$ DNA or 22 pM T4 or T4$_{147}$ genomic DNA in EMSA buffer (0.5× TBE pH 8.0, 35 mM NaCl, 5% glycerol) in a final 12 μL reactions and incubated for 30 min at room temperature. Reactions were run for 2 h on a 1% agarose or 3 h on 0.5% agarose 1× TAE gels at 90 V at 4 °C for $\lambda_{ts}$ and T4 genomes correspondingly.

## Strep-Seq analysis of BrxX binding to T7 DNA
To detect in vivo BrxX methyltransferase binding to the phage DNA in a course of infection, a variant of the ChIP-Seq procedure was developed. Cells carrying pBREX AL BrxX C-Strep were grown in 100 mL LB at 37 °C till $OD_{600} = 0.5$. 40 mL aliquot was taken and mixed with the phage T7$_{fusion}$[40] at MOI = 1, infection proceeded for 15 min. Formaldehyde was added to the culture at a final concentration of 1% and cross-linking was carried out for 5 min at room temperature. Formaldehyde was quenched by adding glycine to a final 500 mM concentration and the mixture was agitated at 4 °C for 20 min. Cells were collected by centrifugation (6000 × $g$, 10 min, 4 °C) and kept at −20 °C overnight. Cells were resuspended in lysis solution (StrepA buffer supplemented with 300 μg/mL lysozyme) and incubated at 37 °C for 30 min following sonication disruption (65% power, 10 s pulse, 20 s pause, 12 cycles on Qsonica sonicator with 2 mm sonotrode). Sonication conditions were optimized to produce DNA fragments of 200–600 bp mean shear size. Samples were treated with RNAse A and 1/20 volume was taken for the purification of total DNA which served as mock DNA control without enrichment. The rest of the sample was loaded on the StrepTrap HP chromatography column to purify Strep-tagged BrxX. The eluted fraction was kept at 65 °C to reverse formaldehyde cross-links. DNA was purified with phenol:chloroform method and sequenced on MiniSeq platform (Illumina) with paired-end 150 cycles (75 + 75). Reads from the StrepTrap enriched sample and mock DNA sample were mapped to the T7$_{fusion}$ genome using bwa mem. Genome coverage was calculated and analyzed with bedtools, genomecov, and samtools and was further normalized to the sequencing depth and the genome size to achieve CPM values. Coverage was visualized using the seaborn package in Python as the ratio of the enrichment to the mock sample CPM values.

## DNA methylation analysis with Oxford Nanopore (ONT)
To identify DNA specificity of the BREX mutants, we performed ONT sequencing, followed by the detection of methylation motifs. Genomic DNA of BW25113 carrying pBREX AL, pBREX AL BrxX$^{K681A/D684S}$, or pBREX AL BrxX$^{G785T/N823S}$ was purified from 2 mL of overnight cultures grown in LB at 37 °C with Monarch Genomic DNA Purification Kit (NEB). Total DNA libraries were prepared from DNA treated with XbaI (Thermo Scientific) using the Native Barcoding Kit 24 V14 (SQK-NBD114-24) with enrichment of long fragments using the Long Fragment Buffer according to the manufacturer's instructions. DNA library was sequenced using R10.4.1 flow cell (FLO-MIN114) on the MinION device with MinKNOW v23.11.2. The basecalling was performed with Dorado v.0.53 (https://

github.com/nanoporetech/dorado) using the following models: a main basecalling model dna_r10.4.1_e8.2_400bps_sup@v4.3.0 and the modified basecalling model dna_r10.4.1_e8.2_400bps_sup@v4.3.0_6mA@v2 (for N6-methyladenosine (6 mA) modifications) from Rerio v. 5.0.11 basecalling models set (https://github.com/nanoporetech/rerio). The resulting basecalls were mapped to *E. coli* BW25113 and pBREX AL plasmid reference according to MicrobeMod pre-processing tutorial[65]. We additionally verified that pBREX AL plasmids lacked any non-specific mutations. Finally, the search of 6 mA methylation motifs was performed with MicrobeMod v.1.0.3[65] with the following parameters: minimal strand coverage–10, minimal modkit methylation confidence–0.66 (modkit v0.2.4, https://github.com/nanoporetech/modkit), minimal percent of methylated reads–0.66, and minimal percent of methylated reads to pass motif to STREME–0.8 (STREME v5.5.2)[66]. The coordinates of all found sites corresponding to found motifs were extracted with seqkit locate v.2.8.0[67], and genomic intervals of motifs and found modifications were intersected with bioframe v.0.5.0[68] (python v. 3.10.10). The resulting distributions of modification fractions for each site were visualized with tidyverse v.2.0.0[69] and ggplot2 v.3.4.3 package (https://ggplot2.tidyverse.org/) of R v.4.2.3.

## Biolayer interferometry (BLI)

Oligos used in BLI are listed in Supplementary Table S3. To anneal dsDNA duplexes, equal amounts of 100 μM 5′-biotinylated forward strand and unlabeled reverse strand oligonucleotides (Syntol) were mixed in the annealing buffer (10 mM Tris-HCl pH 8.0, 50 mM NaCl). After heating to 95 °C for 10 min, the mixture was allowed to slowly cool to room temperature. All kinetic assays with Octet R2 were set up in 96-well plate format using 200 μL reaction volumes and were performed in running buffer containing 20 mM Tris-HCl pH 8.0, 100 mM NaCl, 0.05 mg/mL BSA, and 0.02% Tween-20 at 30 °C with the orbital shake speed of 1000 rpm. Reactions were supplemented with 0.5 mM SAM in appropriate measurement series.

Kinetic assays were performed by first capturing 225 nM of 5′ biotinylated dsDNA (20 or 40 bp DNA containing either the BREX site or polyC sequence) onto SAX Octet biosensor in running buffer. The initial baseline was generated for 60 s, loading of the bait was performed for 100 s, and a second baseline was generated for 150 s. Association with BrxX analyte (0.05–1.6 μM) was performed for 150 s and final dissociation was performed for 450 s. For Ocr competitive binding assay, the initial baseline was generated for 30 s, loading of 40 bp DNA with one BREX site was performed for 100 s, a second baseline was generated for further 150 s, and association was performed for 7 min in wells filled with 1.5 μM BrxX analyte supplemented with either untagged 40 bp dsDNA or Ocr, followed by further 21 min dissociation in running buffer. Ocr molar concentration was calculated for the dimer. In each run, one of the SAX sensors loaded with biotinylated dsDNA was incubated in a running buffer without the analyte (BrxX) to capture the reference signal. It was also validated that incubation of the SAX sensor with BrxX does not result in a binding signal. Fresh SAX biosensors were used without any regeneration step.

The raw data measured on Octet R2 was analyzed using Octet Analysis Studio software (version 12.2.0.20). After single reference subtraction, binding sensorgrams were first aligned at the beginning of the association cycle, inter-step corrected to the association step, and filtered according to the Savitzky-Golay algorithm. Binding sensorgrams were globally fit to a 1:1 Langmuir binding model for 20 bp substrates or to 2:1 heterogenous binding model for 40 bp substrates. Some curves with low protein concentrations were excluded from the analysis due to the lack of response signal.

## Cryogenic electron microscopy (cryoEM) sample preparation

BrxX-DNA complex was prepared containing 250 μM DNA (43 bp, H2T), 50 μM BrxX monomer, and 1 mM SAM. The mixture was dialyzed overnight at 4 °C into cryoEM buffer (20 mM HEPES-NaOH pH 7.5,

2.5 mM MgOAc, 150 mM NaCl, and 0.5 mM TCEP) containing 50 μM SAM. For BrxX-Ocr, 10 mg/mL of reconstituted BrxX-Ocr complex was dialyzed overnight at 4 °C into cryoEM buffer. Following dialysis, SAM was added to 1 mM and CHAPSO to 8 mM final concentrations. Both complexes were spun for 1 h at 15,900 × g prior to grid preparation to remove any potential aggregates. Aliquots of 3.5 μL of BrxX:DNA or BrxX/Ocr complexes were applied to glow-discharged (Leica, 60 s/ 8 mA) Quantifoil holey carbon grids (R2/1, 300 copper mesh). After 30 s of incubation with 100% chamber humidity at 4 °C, the grids were blotted for 3.5 s (BrxX-Ocr) or 4.5 s (BrxX-DNA) and plunge-frozen in liquid ethane using a Vitrobot mark IV (FEI).

## Single-particle cryoEM data collection

**Initial screening and analysis.** For initial screening, data were collected on Talos microscope (Thermo) equipped with a Gatan Elsa side-entry holder & Falcon 4i direct electron detector operated at 200 kV at JIC BioImaging facility. Datasets of 4000–5000 movies in EER mode were collected and analyzed before progressing to Krios microscope data collection.

**BrxX:DNA.** CryoEM data were collected at Astbury Biostructure Laboratory (ABSL) CryoEM facility (University of Leeds, UK) on Krios G2 microscope (Thermo) operated at 300 kV and nominal magnification of 165 kx. Movies were recorded in counting mode on a Falcon 4i direct electron detector (Thermo) in EER format using EPU v 3.6. Movie frames were collected at the calibrated physical pixel size of 0.74 Å/px with a defocus range set to −3 to −0.9 μm. A dose rate and exposure time were chosen to result in a total dose of ~50 electrons/Å². Full statistics for cryoEM data collection are listed in Supplementary Table S4.

**BrxX:Ocr.** CryoEM data were collected at ABSL CryoEM facility (University of Leeds, UK) on Krios G2 microscope (Thermo) operated at 300 kV and nominal magnification of 120 kx. Movies were recorded in counting mode on a Falcon 4i direct electron detector (Thermo) in EER format using EPU v 3.3. Movie frames were collected at the calibrated physical pixel size of 0.68 Å/px with a defocus range set to −3 to −0.9 μm. A dose rate and exposure time was chosen to result in a total dose of ~35 electrons/Å². Full statistics for cryoEM data collection are listed in Supplementary Table S4.

## CryoEM data analysis

**BrxX:DNA.** All processing was done in cryoSPARC[70] v.4.4+. 11 152 movies were motion and CTF corrected in patch mode. After manual curation, 9980 movies were retained for further analysis. Particles were picked with Topaz[71] (trained on a small set of manually-picked micrographs) and 2 × 2 binned particles (610 100) were subjected to a single round of 2D classification (100 classes, 40 iterations, 20 full iterations, batch size 300, max resolution 5). Two lakh sixty-five thousand two hundred ninety-five retained particles displaying high-resolution features underwent 3D classification with 2 classes (ab initio, 219 641 particles retained) and were refined to 3.03 Å resolution. Particles were re-extracted at the original pixel size using updated coordinates. Non-uniform refinement[72] of this particle set with local CTF correction resulted in a 2.47 Å map. To further improve the resolution, particles underwent local reference-based motion correction ("Bayesian polishing") as implemented in cryoSPARC v.4.4+[73]. Two rounds of particle polishing were carried out followed by non-uniform refinement with local defocus refinement, global CTF refinement (tilt, trefoil, spherical aberration, and anisotropic magnification[74]), and Ewald sphere correction[75]; this resulted in a final map with 2.2 Å global resolution as measured by cryoSPARC.

**BrxX:Ocr.** Processing was done in cryoSPARC v 4.1+. Five thousand one hundred collected movies were motion and CTF corrected and

manually curated. Four thousand six hundred seventy-eight movies were retained for further analysis. Twenty-four micrographs were picked manually and used to train Topaz. Topaz-picked particles (244,977) were used to create an initial map of BrxX$_2$:Ocr$_2$ complex. This map was used to create equally-spaced 2D projections to use as templates, and particles were more thoroughly picked using cryosparc template picker (396 923) and extracted as twice binned (1.36 Å/pix). Based on manual analysis of 2D class averages, 227,907 particles were selected that represented a BrxX$_2$:Ocr$_2$ complex with a minor set of particles assigned to BrxX apo (38,496, see below). For 227,907 particles (main set), ab initio 3D classification was used to further separate low-resolution particles. Two lakh eleven thousand eight hundred eighteen particles representing complete complex were retained, and by means of heterogenous refinement (batch size 9990, final iterations 20) separated into two classes of almost equal size (114,826 and 96,992 particles) that differed by the slight change in the orientation of the "upper jaw" of the better-resolved BrxX monomer (see Supplementary Fig. S6). A larger class was retained for more detailed analysis. After a NU-refinement, these particles were classified without alignment using a mask around the "upper jaw" (MTD + NTD). Parameters for the 3D classification job were: 5 classes, 10000 batch size, 20 O-EM epochs, target resolution 4 Å. Four best classes were retained (94,764 particles). Particles underwent one round of Bayesian polishing as implemented in cryoSPARC, followed by local CTF refinement, global CTF refinement (tilt, trefoil, spherical aberration, and anisotropic magnification), and Ewald sphere correction. Final consensus refinement resulted in a 2.91 Å map with the resolution being highest for the Ocr dimer and two BrxX TRDs. To improve the resolution around the "upper jaw" (MTD + NTD), local refinement was carried out using the corresponding mask, resulting in the 2.84 Å focussed map for the MTD + NTD of a single BrxX monomer within the complex. For the particles representing BrxX apo, ab initio 3D classification was carried out with 2 classes. The best class (24,284) was refined to 3.88 Å.

### Model building and refinement
**BrxX:DNA.** AlphaFold2-generated model for BrxX was used as a starting point. The model was split into four separate domains that were first manually rigid-body fitted in ChimeraX[76], followed by manual re-building in Coot[77] as necessary. DNA initial model was built using cryoREAD server[78], followed by manual adjustment in Coot using local restraints. Real-space refinement was performed in phenix.refine[79] (using Ramachandran restrains, and secondary structure restraints for protein and DNA). *BrxX:Ocr*: A refined BrxX:DNA model was split into two halves corresponding to the "upper" and "lower" jaws (residues 1–660 and 661–1205) which were rigid-body fit into the density for the best-resolved monomer. An Ocr-inhibited *E. coli* RNAP structure (PDB: 6R9G) was used to generate an Ocr initial model which was rigid-body fit into the density. Missing elements were manually built in Coot; Ramachandran restraints were used to build loops with weaker density, particularly residues 422–430. After building the best-resolved BrxX monomer, it was copied and rigid-body fitted into the density for the second monomer, using a well-resolved two-helix bundle at the tip of the CTD for guidance. NCS restraints were used in Phenix.refine to stabilize first rounds of refinement, and were switched off during later rounds as the structure is not fully symmetrical. Ramachandran restraints, and secondary structure restraints were used. *BrxX apo*: a well-resolved BrxX monomer from the BrxX:Ocr structure was rigid-body fit in the map, and refined in phenix.refine using Ramachandran and secondary structure restraints. MolProbity[80] and MTriage[81] were used to validate all models and maps. Statistics for the final models are reported in Supplementary Table S4.

### BREX methylation restriction-sensitivity assay
Plasmid pHERD30t with and without overlapping AluI/BREX site was purified from BREX$^+$ (*E. coli* BW25113 + pBREX AL) or BREX$^-$ cells (*E. coli*

BW25113 + pBTB) using Monarch Plasmid Purification Kit (NEB). 240 ng of plasmid was restricted with AluI (Thermo Scientific) according to the manufacturer's instructions. Non-cleaved and cleaved plasmids were run at 1% agarose gel in 1x TAE buffer and stained with ethidium bromide.

### In vitro methylation assay
BrxX or Dam protein (100 µM final concentration) was mixed with 5 µg of the 40 bp dsDNA substrate, containing overlapping BREX/AluI site and a single Dam site, in methylation buffer (20 mM Tris-HCl pH 7.5, 100 mM NaCl, 0.1 mg/mL BSA, 1 mM DTT) optionally supplemented with 2 mM SAM. The reaction was incubated overnight at room temperature. DNA was purified with phenol-chloroform, skipping the 70% ethanol washing step[64]. 350 ng of purified DNA was restricted (according to manufacturer's instructions) with DpnI (NEB) or DpnII (NEB) to monitor Dam site methylation status or with AluI (Thermo-Fisher) to monitor BREX site methylation status. The resulting fragments were run on 12% polyacrylamide gels (37.5:1) in 1× TAE buffer. The gels were visualized with SYBR Gold (ThermoFisher).

### HPLC-MS analysis of nucleosides
One microgram of purified dsDNA substrate incubated with Dam or BrxX, as described above, was digested with Nucleoside Digestion Mix (NEB) at 37 °C overnight. Nucleosides were loaded onto Agilent Poroshell 120 SB-C18 column (4.6 × 100 mm, 2.7 µm) and were analyzed on Agilent 1200 HPLC-MS system with ESI source and Q-TOF detector (Agilent). Gradient conditions were as follows: solution A−5 mM ammonium acetate, pH = 5.3; solution B−90% acetonitrile; LC run was carried at 40 °C, 0.3 mL/min speed and 1 µL of the sample was loaded. The column was washed for 5 min with 2% B, followed by linear increase to 30% B till 30 min, linear increase to 100% B till 36 min, and linear decrease to 2% B till 40 min. UV detection was carried out at 260 nM. LC-MS/MS data were analyzed in MassHunter, nucleosides and their modified variants were searched in EICs of expected m/z values for nucleosides and dm6A.

### Efficiency of plating (EOP) assay
To determine the titer of active phage particles in cell lysates, the double agar overlay method was used. Overnight culture of bacteria (100 µL) was mixed with 10 mL of 0.6% top LB agar supplemented with 0.2% maltose, 5 mM MgSO$_4$, and appropriate antibiotics and poured on the surface of 1.2% LB agar plates. Ten microliter drops of serial 10-fold phage lysate dilutions were spotted on the top agar, allowed to dry and plates were incubated at 37 °C overnight. The level of protection was determined as the ratio of phage titers obtained on a non-restrictive (BREX$^-$) host relative to that on restrictive (BREX$^+$) host. EOP with T4 and T4$_{147}$ phages were conducted on the DH10B host. All experiments were performed in biological triplicates.

### In vivo estimation of BREX methylation via phage λ sensitivity assay
Estimation of the in vivo efficiency of BREX methylation was conducted via EOP assay with the phage λ induced from BREX$^-$, BREX$^+$, or other indicated cultures. The results of this assay correlate with direct estimation of BREX methylation with PacBio sequencing, as shown in our previous works[18,40,43]. Estimation of the pBREX AL BrxX$^{Y511A}$ methylation activity was performed with λ$_{ts}$, as described before[18,40,43], and detailed below. However, λ$_{ts}$ phage was not compatible with the plasmids used for expression of BREX proteins combinations due to the presence of the *Cm$^R$* gene. Therefore, for the determination of the minimal set of proteins required for BREX methylation, we used BW25113 λ$_{wt}$ lysogen. λ$_{wt}$ lysogen was obtained by infecting BW25113 with λ$_{wt}$ in liquid culture and selecting colonies resistant to λ$_{wt}$ superinfection. Lysogenic cells were transformed with pBTB-2 (BREX$^-$ culture), pBREX AL (BREX$^+$ culture), pBAD BrxX, or a set of plasmids expressing BCXZ proteins (pBREX1 ΔBrxA + pBREX2 ΔBrxL)[19]. Overnight cultures were diluted

100× in 35 mL LB with appropriate antibiotics and 0.2% arabinose to induce BREX proteins expression and grown at 37 °C until $OD_{600}$ ~0.4. To induce $\lambda_{wt}$ lysogen, cells were treated with UV. First, to increase the transparency of the culture, cells were harvested by centrifugation (6000 × $g$, 20 °C, 10 min) and washed twice in an equal volume of SM3 buffer (see above) and then in 10 mL SM3 buffer. Suspension of cells was decanted into sterile 100 × 15 mm Petri dish, which was placed without the lid on a shaking platform under the UV lamp (30 W) at a distance of ~50 cm. Suspension was exposed to UV for 30 s. After that, cells were collected in a tube and harvested by centrifugation (6000 × $g$, 20 °C, 10 min). Cell pellet was resuspended in 5 mL LB with 0.2% arabinose and incubated at 37 °C for 3 h until the visible lysis. Cell debris was removed by centrifugation and lysate was cleared with chloroform[82]. The methylation status of collected $\lambda_{wt}$ was measured by determining the ratio of phage titer obtained on a non-restrictive (BREX−) host relative to that on restrictive (BREX+) host.

### Toxicity assay
BREX−, BREX+, or pBREX AL BrxX[Y511A] potential toxicity was estimated in a spot-test assay. Cells were grown in 10 mL of LB and diluted to $OD_{600} = 0.6$, followed by serial 10-fold dilution droplets plating on 1.2% LB agar supplemented with required antibiotics.

### In vivo protein pull-down and size-exclusion chromatography
Overnight cultures of *E. coli* strains with plasmids encoding Strep-tagged BREX proteins in a context of the full BREX cluster[40] (pBREX AL BrxN C-Strep, where N is any of the BREX proteins) were diluted 100-fold in 3 L of LB media and incubated overnight at 18 °C in Thomson Scientific Ultra Yield flasks with aerated lids (1.5 L per flask). Cells were harvested by centrifugation at 4000 × $g$ at 4 °C for 30 min after reaching $OD_{600}$ ~0.9. Pellets were washed in 50 mL of StrepA buffer (150 mM NaCl, 1 mM EDTA, 5 mM 2-mercaptoethanol, 100 mM Tris-HCl pH 8.0), and resuspended in the same buffer supplemented with a cOmplete Protease Inhibitor Cocktail (Roche) and lysozyme (0.2 mg/mL). Cells were disrupted by sonication on ice, and the lysate was clarified by centrifugation at 21,000 × $g$, 4 °C for 30 min. Strep-tagged proteins were purified on two stacked 5 mL StrepTrap HP (Cytiva) columns, connected to the NGC Chromatography System (BioRad). Protein-containing fractions were concentrated using 3 kDa Amicon centrifugal filter units (Merk) and analyzed by SDS-PAGE. The identity of protein bands was determined by matrix-assisted laser desorption/ionization time-of-flight (MALDI-TOF) mass spectrometry. Samples were prepared with Trypsin Gold (Promega) in accordance with the manufacturer's instructions. Mass spectra were obtained using the rapifleX system (Bruker). The molecular weight of protein complexes was determined by size-exclusion chromatography performed on Superdex 200 Increase 10/300 column (GE Healthcare) calibrated with High Molecular Weight calibration kit (GE Healthcare). Protein fractions were checked for DNA presence by incubation with DNAse I at 37 °C and electrophoresis in 1% agarose 1× TAE gel and ethidium bromide staining.

### Protein stability estimation with a label-free thermal shift analysis
To estimate BrxX and BrxX[Y511A] thermostability, 7 μM of protein was mixed with 0.3 mM of either SAM or SAH or was incubated without co-factors at room temperature for 10 min in a 10 μL buffer (20 mM Tris-HCl pH 8.0, 100 mM NaCl). The mixture was loaded into Tycho NT.6 capillaries and protein stability was monitored by fluorescence (as a ratio of intrinsic tryptophan to tyrosine fluorescence detected at 350 nm and 330 nm, correspondingly) upon a gradient temperature increase (as a 30 °C/min temperature ramp applied from 35 to 95 °C).

### Quantification and statistical analysis
All toxicity and EOP assays were performed at least in three biological replicates. All values are presented in the form of a mean +/- standard error. EMSAs and in vitro reactions with AluI-containing dsDNA substrates were performed in triplicates and representative EMSA gels and HPLC profiles are presented. All calculations were performed in MS Excel.

### Reporting summary
Further information on research design is available in the Nature Portfolio Reporting Summary linked to this article.

## Data availability
All data needed to evaluate the conclusions in the paper are present in the paper and/or the Supplementary Materials. Source data are provided as a Source Data file. NGS sequencing data generated in this study have been deposited in the NCBI database under accession code PRJNA1077651. The atomic models coordinates for BrxX:DNA, BrxX:Ocr and apo BrxX have been submitted to the Protein Data Bank (https://www.rcsb.org/) with PDB IDs 9EWZ, 9EX7, and 9EXH, respectively. Corresponding EM maps have been submitted to the Electron Microscopy Data Bank (https://www.ebi.ac.uk/pdbe/emdb/) with IDs EMD-50027, EMD-50029, EEMD-50032, EMD-50028, and EMD-50038. Raw data were submitted to the Electron Microscopy Public Image Archive (https://www.ebi.ac.uk/pdbe/emdb/empiar/) with IDs EMPIAR-12102 and EMPIAR-12103. Source data are provided with this paper.

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

## Acknowledgements

Skoltech team was supported by the RSF grants (22-14-00004, 24-74-10089, and 22-74-00126). We thank Olga Musharova and Ekaterina Savitskaya who sadly passed away, for help with Strep-Seq sequencing, Alina Demkina for help with plasmids sequencing, Polina Zhurlova for assistance with cloning, and Evgeny Klimuk for help with protein purification. Mass-spectra were obtained at Skoltech Advanced Mass-Spectrometry Core Facility with support of the internal Skoltech grant. K.S. and A.T. are supported by RSF grant (24-14-00181). D.G. is a recipient of a Sir Henry Dale Fellowship (221868/Z/20/Z) funded jointly by the Royal Society and Wellcome Trust; work in his lab is also supported by the BBSRC-funded Institute Strategic Programme "Harnessing Biosynthesis for Sustainable Food and Health" (HBio) (grant number BB/X01097X/1). CryoEM data were collected in the Astbury Biostructure Laboratory (ABSL) CryoEM facility; we thank all staff, particularly Yehuda Halfon for excellent technical support. Two hundred kilovolt test cryoEM data were collected at John Innes Centre BioImaging facility with the help of Jake Richardson. T.R.B. was supported by the Royal Society International Exchange Grant (IEC \R2\202085), and a Lister Institute Prize Fellowship. K.L.M. is the Canada Research Chair in Bacteriophage Biology and Therapeutics (CRC-2023-00010) and is supported by the Canadian Institutes of Health Research (PJT-165936).

## Author contributions

A.I., K.S. and D.G. initiated the study, A.D., M.S., K.P., D.Y. and A.I. performed biochemistry experiments, M.C.A. prepared cryoEM samples, collected, and analyzed cryoEM data. M.C.A. built and analyzed atomic models. M.M. constructed Strep-tagged plasmids and contributed to the in vivo pull-downs. A.T. performed ONT sequencing, and O.K. analyzed ONT data. A.I., D.G., K.S. and T.R.B. acquired funding. A.I., D.G., K.S., T.R.B. and K.L.M. provided resources for this study. A.I., M.C.A., A.D., M.S. and D.G. prepared figures and the initial manuscript draft. A.I., D.G. and K.S. edited the manuscript with comments from T.R.B. and K.L.M. All authors have read and agreed with the final version.

## Competing interests

The authors declare no competing interests.
