## [Transparent Peer Review file · Nature Communications]

Molecular basis of foreign DNA recognition by BREX anti-phage immunity system

Corresponding Author: Dr Artem Isaev

Version 0:

Reviewer comments:

Reviewer #1

(Remarks to the Author)

In this manuscript, the authors present an exciting new contribution to the BREX field, providing insight into how the methyltransferase BrxX recognizes and methylates target DNA, how this enzyme is inhibited by Ocr, how BrxX may be engineered to alter site specificity, and into what BREX components are required for methylation activity. This is a very important step forward to gaining step-by-step mechanistic insight into the function of BREX, which has remained enigmatic since its discovery. Below I suggest a small number of additional experiments and text changes that I feel would clarify and strengthen their manuscript. Thank you for sharing your exciting work.

Major Points:

- The authors make claims in the text and figures which are not supported by their data:

1. Line 97–99, lines 109–142: The authors claim they demonstrate that “BREX response is based on foreign DNA recognition by BrxX.” The authors compellingly show that BrxX is the only protein, when expressed/purified alone, that specifically recognizes BREX sites (Fig S1A), and that it binds to such sites in vitro (Fig 1C and 1F) and in phage DNA during infection (Fig 1D). However, is it not possible that restriction (however this occurs) in response to foreign sites proceeds by an alternative mechanism that is not gated by BrxX? Couldn't BrxX just be binding to the unmethylated phage DNA? How do we know it is mediating the downstream response from these experiments? Without being able to explain (with data) how BrxX/the BREX complex discriminates self- from non-self DNA and how it activates downstream activity of the BREX complex, it would seem premature to dub BrxX a sensor.

2. Figure 7: I find Panels 7A and 7B to be a bit distracting from the story presented in this manuscript. A lot of interesting things were discovered in Figures 1–6. Why present an idea for self/nonself discrimination here which is not investigated in the paper? I suggest the authors create a unified, single panel model figure summarizing their results rather than attempting to explain how self/nonself discrimination may work.

- Figure 3D/E: Altered specificity via engineering of this BREX construct is a very exciting result which could be better supported by the data. No ONT methylation base calling data is presented for the WT parent strain, which is an essential control for evaluating how these traces have changed. It would also be useful to include an analysis for each mutant similar to panel 5D incorporating defense phenotypes for lambda and T7, lambda ts compatibility data, and viability data.

- Figure 5E: In this positive control experiment for using AluI to read out methylation status of their plasmid DNA, the authors used plasmid derived from the E. coli HS strain, calling it “BREX+” and compare it to BW25113 or “BREX-“ in this figure. Given that these strains are not isogenic, it is conceivable that E. coli HS expresses other RM systems or methyltransferases which could overlap and block AluI restriction in a BREX-independent manner. Suggest performing this assay again using plasmids purified from an otherwise isogenic pair of strains that are BREX+ and BREX- (e.g. using the BW25113-derived strain called “BREX+” in the methods, or an E. coli HS strain which is a BREX knockout). Given that another paper (<https://academic.oup.com/nar/advance-article/doi/10.1093/nar/gkaf608/7710916>) was recently published used a different (also indirect) method of measuring methylation and did observe in vitro activity of BrxX alone, this result (positive or negative) is important.

Minor Points:

- Figure 1D: Some peaks do not appear to correspond to BREX sites at all, say around nucleotide 33,000 and at the termini of the genome. Is there an explanation for this? Was any host DNA mapped?
- Figure 1 and Figure S1: Please provide lengths of all oligos so we can better understand the data.
- Figure S1: Typo – 0.75 μ M should be 0.075, as it is under 0.1 μ M.
- The metal binding site observed in the structure is interesting and unexpected. The authors assert it is conserved, but do not provide a sequence alignment to back this up, or perform mutations to assess its importance for BREX defense.

James Eaglesham

Reviewer #2

(Remarks to the Author)

The BREX (Bacteriophage EXclusion) system employs epigenetic DNA methylation to differentiate between host and invading DNA. In Type I BREX systems, this defense relies on the BrxX (PglX) methyltransferase and the S-adenosyl methionine (SAM) cofactor for site-specific DNA recognition and modification. A detailed 2.2-Å cryoEM structure of *Escherichia coli* BrxX bound to its target DNA was presented in this paper. However, BrxX alone is insufficient for methylation; the authors showed that a complex assembly of BrxBCXZ is necessary. Additionally, the authors present a cryoEM structure of BrxX bound to the phage-encoded inhibitor Ocr which reveals that Ocr sequesters BrxX in an inactive dimeric form. A convincing model is presented where BREX-mediated exclusion of phage DNA involves a multi-subunit BrxBCXZ complex and detailed DNA recognition by BrxX. Overall, this work advances our understanding of BREX mechanisms and hints at the potential for bioengineering applications.

The authors present a thorough and well-detailed investigation into the BREX with a focus on the role of the BrxBCXZ complex in DNA methylation and defense against phages. The study effectively highlights the importance of BrxX as a sensor for BREX sites, demonstrating its dual role in both the recognition of viral DNA and initiation of defense mechanisms. Future research includes validation of their model of BREX defense to confirm the proposed mechanisms, particularly the role of BrxC and BrxL in DNA translocation and defense activation, respectively. Overall, the research provides significant insights into the BREX system. The authors should be commended on providing such a clear story that includes both detailed methodologies and highly effective visuals. Minor suggestions for improvement are shared below.

- The size estimates for some of the oligomeric complexes (Fig. 6) are possibly off given that a S200 column was used for chromatographic separation which does not have great resolving power above 200 kDa (near the void/dead volume). If possible, a Superose-6 or some other column would be a better way to establish the size estimates. SEC-MALS, AUC, or DLS are other alternatives.
- “air quotes” are perhaps misused or overused throughout the manuscript. A primary example would be- “local density” (line 70) where the use of the quotations is unnecessary.
- BREX should likely be defined upon first use outside of the abstract (line 56).
- To maintain consistency throughout, numbers one through nine should be spelled out in words (line 58 write out six instead of “6”, line 339, line 751)
 - o Other examples: Line 331,343,344: write out “2nd and 4th” vs. 336: “fourth C:G pair”
- Inconsistency- Line 127 “43 bp”; Line 164 “43-bp”; Line 216 “25 base pairs”
- Line 209- missing a comma between “11” and “152”?
- Define EOP (line 323) and ONT (line 325) in main text upon first use (rather than rely on definition in the methods)
- Line 549- unnecessary dash between “BrxX” and “with BrxC”.
- Line 790, 793, 955- missing commas
- Line 818- define “mQ”
- Line 910- missing degree symbol

Reviewer #3

(Remarks to the Author)

Reviewer #4

(Remarks to the Author)

The BREX system protects against phage infection by methylating DNA to discriminate between host and invading DNA. However, the mechanism by which BREX systems recognize foreign DNA is currently unknown. In the current work, the authors report the cryoEM structures of *Escherichia coli* BrxX bound to target dsDNA and BrxX bound to the phage-encoded inhibitor Ocr. Based on these structures and various biochemical experiments, the authors propose a mechanism for BREX defense and recognition (and inhibition) of invading DNA.

I have some major comments that I suggest to address, as well as various minor comments. I hope that the authors find my comments useful and that they can improve their findings.

Major comments

1. The mechanism by which BrxX discriminates between host and invading DNA remains vague and speculative (or at least the description is unclear to me).
2. The authors describe key residues important for DNA binding, recognition and Ocr binding, but there is no experimental validation of most of these interactions. Point mutants should be made and constructed to verify whether the residues described are important for DNA binding, recognition and Ocr binding. This is essential to validate their structural observations and to strengthen their conclusions.

Minor comments:

1. Line 169: "decreasing dsDNA length resulted in weaker binding", can this observation be explained by the structure?
2. Line 211: "the presence of DNA downstream of a BREX site is essential for BrxX binding", can this observation be explained by the structure?
3. Line 247: N1012, N997, N999 do not correspond to the labels in Fig. 2D.
4. The author should label more information in the figures, for example, the authors cite Fig. 2D when they mention the flexible hinge loops in line 239, but in fact the loops are not labeled in Fig. 2D. and the sequence information should be shown in Fig. 2E.

Version 1:

Reviewer comments:

Reviewer #1

(Remarks to the Author)

I thank the authors for their thoughtful responses to my queries and those of the other reviewers. I enjoyed reading the exciting additions to their manuscript. All of my major points have now been addressed and I think the manuscript should be published without further revision. Thank you for sharing your work.

James Eaglesham

Reviewer #4

(Remarks to the Author)

Dear editor, dear authors,

While the authors have addressed some of the issues that I have raised in their rebuttal, several of my points remain unaddressed in the manuscript.

1. The authors did not adequately address my question regarding the molecular mechanism by which BrxX discriminates between bacterial and phage DNA. Since the authors have solved the structure of the BrxX-DNA complex, they should provide both structural and functional evidence to provide a clear and reasonable explanation for the selectivity of BrxX for bacterial DNA over phage DNA.

Furthermore, the authors state that "a single BrxBCXZ complex likely combines both activities and relies on BrxX for DNA recognition, methylation of host DNA, and sensing non methylated DNA of invading mobile elements" (Line 586-588). Does this suggest that the BrxBCXZ complex, with its methylation activity, is capable of methylating both host and phage DNA? If, as the authors claim, BrxBCXZ only methylates host DNA, why does it not methylate phage DNA after binding, especially considering it has a stronger binding affinity for phage DNA? How does this selectivity occur, and what happens after BrxBCXZ binds phage DNA? Additionally, how exactly does the BREX system carry out its anti-phage defense mechanism?

2. The authors constructed a series of BrxX mutants targeting key residues important for DNA binding, recognition, and Ocr binding and evaluated their effects. However, there are still many issues that need to be addressed before the manuscript can be considered for publication:

--a. In Table S1, the authors refer to three loop mutations: 593DGAT599, 588EYSGFDGATVPI602, and 597VTA599. However, the authors did not clarify how these loops were mutated - were they deleted or substituted with other residues? This should be explained.

--b. For the mutations in key amino acids involved in DNA binding and BREX site recognition, the authors only evaluated the effect of one of the mutations on DNA binding. For the other mutations, only the effect on BREX defense was measured. However, the effect on BREX defense alone does not functionally validate that these residues are directly involved in DNA binding or site recognition, as the authors claim. It is possible that the loss of defense could simply be due to improper expression or folding of these mutants in bacteria. The authors should evaluate the effect of these mutations on DNA binding or site-specific recognition in vitro, similar to the approach used for K515E/K522E/K526E/K533E. In addition, the effect of these mutations on Ocr binding should also be evaluated for the Ocr-binding mutants.

3. Furthermore, there are many illogicalities in the article that make it difficult to understand this manuscript.

--a. The authors state: "We show that BrxX is the only BREX component that can recognize BREX sites in vitro. BrxX also binds to non-methylated BREX sites in phage DNA in vivo. Thus, BrxX must be involved in viral DNA sensing and the initiation of the BREX defense"(Line 620-623). This conclusion is difficult to follow because it only shows that BrxX is necessary for binding to BREX sites, but does not exclude the possibility that other mechanisms could initiate the BREX defense. In fact, Reviewer 1 raised a similar concern, and the authors' response in the rebuttal letter only suggests that BrxX is important for BREX site recognition and BREX defense. However, there is no evidence that BREX site recognition by BrxX is temporally prior to BREX system activation, nor is there direct evidence that BrxX binding to BREX sites is a direct trigger for BREX system activation.

--b. The authors write: "The most widespread Type I BREX employs an R-M like principle: it modifies cell's own DNA at specific sites, and the acquisition of BREX-specific modification protects infecting phage from exclusion" (Line 61-63). This sentence is unclear - does the acquisition of BREX-specific modification refer to host DNA or phage DNA? The broad generalization of background knowledge in this single sentence may confuse readers unfamiliar with the BREX system. The authors should clarify which DNA is being modified and provide more context for readers.

4. The authors seem to have overlooked my earlier advice to include more information in the figures. The current figures still have problems that distract the reader and contain several errors:

--a. The authors refer to 'a positively charged lining of the upper jaw' consisting of K226, K522, K515, and K533 of MTD (lines 291-292), citing Figures 2F and 3A. However, Figure 2F does not show the positions of these amino acids, and Figure 3A does not display all the mentioned residues.

--b. The authors state that 'In contrast, in *E. fergusonii* BrxX, this loop is markedly shorter and is thus able to accommodate the T:A base pair (see Figure 3C)' (lines 326-327). However, in Figure 3C, *E. fergusonii* BrxX is not shown with a T:A base pair.

--c. In Figure 6A, what is represented in green—DNA or SAM?

--d. The authors mention T82 in Figure 5F (line 420), but there is no T82 in Figure 5F. Should this reference be to T821 instead?

--e. What do the labels 90i, 60i, and 147.2i in Figure 1 mean?

Reviewer #1:

This Reviewer stated that we “present an exciting new contribution to the BREX field, providing insight into how the methyltransferase BrxX recognizes and methylates target DNA, how this enzyme is inhibited by Ocr, how BrxX may be engineered to alter site specificity, and into what BREX components are required for methylation activity”. He further stated that our work “is a very important step forward to gaining step-by-step mechanistic insight into the function of BREX, which has remained enigmatic since its discovery” and suggested “a small number of additional experiments and text changes that I feel would clarify and strengthen their manuscript”.

We thank the Reviewer for positive evaluation of our work. We have addressed the comments as described below. The corresponding changes are highlighted in yellow in the marked version of the manuscript.

Major Points:

- The authors make claims in the text and figures which are not supported by their data:

1. Line 97–99, lines 109–142: The authors claim they demonstrate that “BREX response is based on foreign DNA recognition by BrxX.” The authors compellingly show that BrxX is the only protein, when expressed/purified alone, that specifically recognizes BREX sites (Fig S1A), and that it binds to such sites *in vitro* (Fig 1C and 1F) and in phage DNA during infection (Fig 1D). However, is it not possible that restriction (however this occurs) in response to foreign sites proceeds by an alternative mechanism that is not gated by BrxX? Couldn't BrxX just be binding to the unmethylated phage DNA? How do we know it is mediating the downstream response from these experiments? Without being able to explain (with data) how BrxX/the BREX complex discriminates self- from non-self DNA and how it activates downstream activity of the BREX complex, it would seem premature to dub BrxX a sensor.

We build our logic on the facts mentioned by the Reviewer (BrxX is the only BREX protein specifically binding to BREX sites *in vitro* and is able to bind to the non-methylated BREX sites *in vivo*) and on an additional important argument: deletion of *brxX* or inactivating mutations in the catalytic NPPY motif abolish BREX defense. Since we show that the NPPY motif is essential for recognition of BREX sites, it could be inferred that the inability of BrxX to bind BREX sites abolished the defense (Fig. 6D). This idea is further supported by additional BrxX mutants described in the revised version of the manuscript. The inability of BrxX to recognize BREX sites *in vitro* results in the lack of BREX defense *in vivo* (Fig. 4A), while BrxX mutant with relaxed BREX site specificity increases the level of BREX defense *in vivo* (Fig. 4C,D). These results strongly support the idea that recognition of non-methylated DNA by BrxX initiates the BREX defense and controls its strength. In addition, we show that BrxX is a part of a large BrxBCXZ complex, which provides a possible mechanistic link between recognition of non-methylated sites and the activation of the defense pathway, similar to situation in Type I and III R-M systems.

2. Figure 7: I find Panels 7A and 7B to be a bit distracting from the story presented in this manuscript. A lot of interesting things were discovered in Figures 1–6. Why present an idea for self/nonself discrimination here which is not investigated in the paper? I suggest the authors create a unified, single panel model figure summarizing their results rather than attempting to explain how self/nonself discrimination may work.

We agree that this section of the Discussion may look speculative at this stage of research. The proposed model directly stems from our attempts to explain the obtained results: BrxX is unable to methylate DNA on its own and requires the contribution from other BREX proteins to support methylation process. We updated the Discussion section and shifted the focus away from the self/nonself discrimination question.

- Figure 3D/E: Altered specificity via engineering of this BREX construct is a very exciting result which could be better supported by the data. No ONT methylation base calling data is presented for the WT parent

strain, which is an essential control for evaluating how these traces have changed. It would also be useful to include an analysis for each mutant similar to panel 5D incorporating defense phenotypes for lambda and T7, lambda ts compatibility data, and viability data.

We now provide additional results for TRD-specificity mutants, as well as additional mutants requested by other Reviewer: EOP against λ_{vir} and T7 Δ 0.3, ONT control for WT BREX, as well as *in vitro* EMSA results. We confirm that WT BrxX efficiently binds only to DNA substrates with complete BREX sites. In contrast, BrxX N823S/G785T mutant, which indiscriminately methylates any nucleotide at the fourth position, is able to recognize all four GGTNAG BREX sites, while its overall DNA binding efficiency is decreased. We suppose that the difference between the amount of available BREX recognition motifs and the overall efficiency of DNA binding could explain the observed discrepancy in defense against λ_{vir} and T7 Δ 0.3 phages for the K681A/D684S mutant. These results are summarized in a new **Figure 4** of the revised manuscript.

- Figure 5E: In this positive control experiment for using AluI to read out methylation status of their plasmid DNA, the authors used plasmid derived from the E. coli HS strain, calling it "BREX+" and compare it to BW25113 or "BREX-" in this figure. Given that these strains are not isogenic, it is conceivable that E. coli HS expresses other RM systems or methyltransferases which could overlap and block AluI restriction in a BREX-independent manner. Suggest performing this assay again using plasmids purified from an otherwise isogenic pair of strains that are BREX+ and BREX- (e.g. using the BW25113-derived strain called "BREX+" in the methods, or an E. coli HS strain which is a BREX knockout). Given that another paper (<https://academic.oup.com/nar/advance-article/doi/10.1093/nar/gkae608/7710916>) was recently published used a different (also indirect) method of measuring methylation and did observe *in vitro* activity of BrxX alone, this result (positive or negative) is important.

As requested, we repeated our *in vivo* BREX methylation restriction-sensitivity assay using an isogenic pair of BREX- and BREX+ cells. We made these experiments in both systems (purification of reporter plasmid from pBTB and pBREX AL BW25113 *E. coli* host or from the *E. coli* HS and *E. coli* HS Δ BREX strain) with a consistent result and updated **Figure 6E** with results from one model system. We also were puzzled by the referenced observation, which appeared after our paper was already submitted. The authors of the mentioned paper did not measure the proportion of DNA substrate that was methylated; thus, it is hard to estimate the activity of BrxX, in contrast we measured methylation activity by directly or indirectly monitoring accumulation of methyl-adenines in DNA substrate. It is not excluded that under *in vitro* conditions a small fraction of BrxX acquires methylation-competent conformation, while *in vivo* methylation-competent conformation requires assembly of the BrxBCXZ complex. Another recently published paper that studied the same protein with an *in vitro* methylation sensitivity assay also reported the lack of activity (<https://www.nature.com/articles/s41467-024-51629-7>). We believe that the *in vivo* results provided in our work (**Figure 7A**) represent strong argument that BrxX on its own does not demonstrate biologically relevant methylation activity.

Minor Points:

- Figure 1D: Some peaks do not appear to correspond to BREX sites at all, say around nucleotide 33,000 and at the termini of the genome. Is there an explanation for this? Was any host DNA mapped?

Unlike transcriptional regulators that constantly occupy their DNA binding sites, the BrxX methyltransferase might only transiently interact with BREX sites, making the ChIP-Seq signal noisy. The experiment was performed with minor modifications 4-5 times, and always produced rather low signal-to-noise ratio with additional peaks present. While peaks at certain BREX sites were highly reproducible, peaks like those at 33,000 bp and 4,8000 were sometimes absent. We initially thought that if BREX acts similar to Type I R-M systems, we could expect additional peaks in-between two properly oriented BREX recognition sites, in places of possible collision between two translocating complexes. Alternatively, some non-specific peaks could represent DNA fragments co-purified with other host proteins that non-specifically adsorb to the Streptactin resin. With the current signal-to-noise ratio it is challenging to interpret these results. We plan to

change the BrxX purification system to 3xFLAG-tag and study BrxX distribution on DNA in more detail. Mapping of reads on host genome did not show any signs of enrichment at the BREX sites. We therefore submit that the current Strep-Seq result, which certainly can be improved (and will be improved in the future), provides good support for the recognition of non-methylated BREX sites in phage DNA by BrxX.

- Figure 1 and Figure S1: Please provide lengths of all oligos so we can better understand the data.

We have tried to indicate the length of oligos on all EMSA panels, yet it rather cluttered the Figures. Thus, we prefer not to show these data on the panels themselves. Please note that the requested information is provided in Figure legends for all EMSA panels. **Supplementary Figure S1B** indicates the length only because it is a varied parameter, while on all other panels the varied parameter is the sequence of oligos.

- Figure S1: Typo – 0.75 μ M should be 0.075, as it is under 0.1 μ M.

Thank you, corrected.

- The metal binding site observed in the structure is interesting and unexpected. The authors assert it is conserved, but do not provide a sequence alignment to back this up or perform mutations to assess its importance for BREX defense.

When we examine the metal binding site in the context of our ConSurf analysis, we see that these five residues have the highest conservation score (grade 9) across Type 1 BREX systems. We have created an inset to **Supplementary Figure S5A** to highlight this fact and provided a sequence alignment of the metal binding site among BxrX homologs (**Supplementary Figure S5F**). We attempted to mutate the ion binding site and constructed a BxrX D997A/D999A/D1012A mutant. However, this mutant did not demonstrate compromised BREX defense, and rather showed an enhanced defense (see below). It is possible that mutations of additional residues involved in metal ion coordination are required to prevent ion binding or that in the lack of ion binding BrxX is destabilized which affects the level of BREX defense. We plan to further investigate ion binding with this mutant *in vitro*, yet we suppose that it will constitute a part of the follow-up research.

Reviewer #2:

This Reviewer stated that our data allowed us to propose “a convincing model ... where BREX-mediated exclusion of phage DNA involves a multi-subunit BrxBCXZ complex and detailed DNA recognition by BrxX” and

that “overall, this work advances our understanding of BREX mechanisms and hints at the potential for bioengineering applications”. They commended us “on providing such a clear story that includes both detailed methodologies and highly effective visuals” and offered “minor suggestions for improvement”.

We thank the Reviewer for kind comments and appreciation of our work.

- The size estimates for some of the oligomeric complexes (Fig. 6) are possibly off given that a S200 column was used for chromatographic separation which does not have great resolving power above 200 kDa (near the void/dead volume). If possible, a Superose-6 or some other column would be a better way to establish the size estimates. SEC-MALS, AUC, or DLS are other alternatives.

We have used a Superdex 200 10/300 Increase column from Cytiva which has a reported resolution range between Mr 10.000 – 600.000 and as such might indeed render size estimations of the BrxBCXZ and BrxCX complexes inaccurate. We highlighted this limitation in the manuscript. However, the exact sizes are not critical for this manuscript as no major conclusions are made other than simply highlighting that higher oligomeric complexes are formed. A detailed investigation of the BREX supramolecular complexes and their stoichiometry is ongoing but is beyond the scope of this work.

- “air quotes” are perhaps misused or overused throughout the manuscript. A primary example would be “local density” (line 70) where the use of the quotations is unnecessary.

Thank you, corrected throughout the manuscript.

- BREX should likely be defined upon first use outside of the abstract (line 56).
- To maintain consistency throughout, numbers one through nine should be spelled out in words (line 58 write out six instead of “6”, line 339, line 751)
 - o Other examples: Line 331,343,344: write out “2nd and 4th” vs. 336: “fourth C:G pair”
- Inconsistency- Line 127 “43 bp”; Line 164 “43-bp”; Line 216 “25 base pairs”
- Line 209- missing a comma between “11” and “152”?
- Define EOP (line 323) and ONT (line 325) in main text upon first use (rather than rely on definition in the methods)
- Line 549- unnecessary dash between “BrxX” and “with BrxC”.
- Line 790, 793, 955- missing commas
- Line 818- define “mQ”
- Line 910- missing degree symbol

Thanks for picking these up. We did our best to correct or clarify these and other inconsistencies.

Reviewer #3:

This Reviewer co-reviewed our manuscript with one of the reviewers who provided the listed reports.

Reviewer #4:

The Reviewer had “some major comments that I suggest to address, as well as various minor comments” to improve our findings. We addressed these comments as is detailed below.

Major comments

1. The mechanism by which BrxX discriminates between host and invading DNA remains vague and speculative (or at least the description is unclear to me).

This is now addressed (see also our response to Reviewer 1). We have re-written and simplified this section, focussing on BrxX and removing some more speculative models as requested. Hopefully the Reviewer finds that the amended text is improved.

2. The authors describe key residues important for DNA binding, recognition and Ocr binding, but there is no experimental validation of most of these interactions. Point mutants should be made and constructed to verify whether the residues described are important for DNA binding, recognition and Ocr binding. This is essential to validate their structural observations and to strengthen their conclusions.

As requested, we have constructed a series of BrxX mutants and evaluated the effects of amino acid substitutions on BREX defense and methylation specificity *in vivo*, and on DNA binding and BREX sites recognition by BrxX *in vitro*. Below is the list of all the new mutants tested in the revised manuscript (**Supplementary Table S1**). Most of the new results are now presented in new **Figure 4**, while EOP data with Ocr-binding mutants are presented in updated **Figure 5**.

Predicted role of mutated residues	Mutation	Defense	DNA binding	Other features
Non-specific DNA binding (backbone interactions)	K515E/K522E/K526E/K533E	-	-	
	K1076E/K1077E	-		
	K719E	+		
Recognition of the 5th position in BREX site via a loop in MTD	593DGAT599	-		
	588EYSGFDGATVPI602	-		
	597VTA599	-		
Changing the specificity of BrxX TRD at the 2nd BREX site position G->C (E. fergusonii as reference)	K681E/D684T	-		
Changing the specificity of BrxX TRD at the 2nd BREX site position G->A (S. typhimurium as reference)	K681A/D684S	+		Modified BREX specificity
	679IALKAGMST688	-		
Changing the specificity of BrxX TRD at the 4th BREX site position A->C (S. typhimurium as reference)	G785T/N823S	++	+	Modified BREX specificity
	N823S/G785T + 784DTGR787	-		
Changing the specificity of BrxX TRD at the 1st and the 3rd BREX site positions	Q784N/R751Q	-		
Ocr binding	G1138A/T1141A/R1145E	+		T7 phage sensitive
	G1138A/T1141A/R1145E/K1048D	+	+/-	T7 phage resistant, reduced DNA binding
Metal ion binding	D997A/D999A/D1012A	+		
Methyltransferase catalytic motif	Y511A	-	+/-	Non-specific DNA binding

In short, mutations affecting backbone DNA interactions result in the loss of phage defense *in vivo* and BREX sites recognition *in vitro*. Mutations in the TRD have pleiotropic effects, ranging from the loss of phage defense to changes in BREX site specificity. The G785T/N823S mutant demonstrates enhanced BREX defense levels, consistent with its relaxed specificity. We constructed a mutant that was likely compromised in Ocr binding. Although this mutant demonstrates decreased DNA binding *in vitro*, it allows BREX to protect cells from wild-type T7 phage that expresses Ocr and overcomes wild-type BREX defense. We thank the Reviewer for suggesting these experiments, they certainly made our work stronger.

Minor comments:

1. Line 169: "decreasing dsDNA length resulted in weaker binding", can this observation be explained by the structure?

Similar observations were made following the cryoEM work. We have originally attempted to use a 14 bp and 20 bp DNA substrates based on EMSA data. However, we were unable to generate any 2D classes of BrxX bound to DNA. This implies that the binding was too weak to produce stable complexes on cryoEM grids. From our final structure, with the 43 bp substrate, we can see why increasing the dsDNA length is necessary (see also the next comment). The DNA is significantly distorted (highlighted in **Figure 2E**) to be able to fit within the BrxX molecule. Extended length and interactions with the CTD (which should also include unmodelled DNA), help compensate for destabilization of the flipped adenine and secure the DNA to eventually facilitate methylation.

2. Line 211: "the presence of DNA downstream of a BREX site is essential for BrxX binding", can this observation be explained by the structure?

Yes, this observation can be explained by the structure. Given the failure of BrxX to bind DNA shorter than 43 bp, we believe that a longer substrate is required to stabilize the distortion caused by the flipped adenine. Although BrxX is able to bind the DNA mimic Ocr and other dsDNA non-specifically, to bind specific sites and enable adenine methylation, longer DNA molecules capable of interacting with the CTD are required. In **Figure 2F**, we note that the entire lower 'jaw' of BrxX (the TRD & CTD) as well as the hinge region (between the MTD and TRD) are positively charged which would distort the DNA molecule and position it at the required angle for the flipped adenine to be inserted into the methyltransferase domain. We have amended the description to clarify this point:

"29 out of 35 direct interactions between the BrxX monomer and DNA, occur within the six bases involved in site-specific recognition. The distorted state of DNA required for site-specific recognition is stabilized by a series of positively charged residues forming direct contacts with the phosphate backbone. A positively charged lining of the 'upper jaw' consists of K226, K522, K515 and K533 of MTD (Figure 2F and Figure

3A); the complementary positively charged groove in the 'bottom jaw' includes K719 and R944 of the TRD, and R1145 and K1152 of the CTD that stabilize the bent 3' tail of the modelled DNA segment. This 3' tail of the DNA fragment outside of the recognition site is essential for stable binding and lies within a positively charged groove of the BrxX-specific CTD (**Figure 2F**). The overall shape and mode of DNA binding resembles the one observed in Mmel:DNA complexes (**Supplementary Figure S4D**)."

3. Line 247: N1012, N997, N999 do not correspond to the labels in Fig. 2D.

We thank the reviewer for catching this. The residues in the text have now been adjusted to correspond to labels in Figure 2D. The corresponding sentence now reads:

"A metal ion is coordinated by five residues: side chains of D997, D999, D1012, Y1113, and the main chain oxygen of I1001, with the last valence occupied by a water molecule."

4. The author should label more information in the figures, for example, the authors cite Fig. 2D when they mention the flexible hinge loops in line 239, but in fact the loops are not labeled in Fig. 2D. and the sequence information should be shown in Fig. 2E.

We have now adjusted the inset of **Figure 2D** to better highlight the flexible hinge loops. The DNA sequence information is shown in **Figure 2A**. **Figure 2E** shows divergence of the shape of DNA bound to BrxX (white) from idealised B-form DNA (black). We have adjusted the figure legend to more clearly convey this.

Response to additional comments by Reviewer 4

1. The authors did not adequately address my question regarding the molecular mechanism by which BrxX discriminates between bacterial and phage DNA. Since the authors have solved the structure of the BrxX-DNA complex, they should provide both structural and functional evidence to provide a clear and reasonable explanation for the selectivity of BrxX for bacterial DNA over phage DNA.

Furthermore, the authors state that “a single BrxBCXZ complex likely combines both activities and relies on BrxX for DNA recognition, methylation of host DNA, and sensing non methylated DNA of invading mobile elements” (Line 586-588). Does this suggest that the BrxBCXZ complex, with its methylation activity, is capable of methylating both host and phage DNA? If, as the authors claim, BrxBCXZ only methylates host DNA, why does it not methylate phage DNA after binding, especially considering it has a stronger binding affinity for phage DNA? How does this selectivity occur, and what happens after BrxBCXZ binds phage DNA? Additionally, how exactly does the BREX system carry out its anti-phage defense mechanism?

We apologize if the text was not clear enough. Like any other restriction-modification system, BREX relies on methylation status to discriminate between host (methylated, thus protected) and foreign (unmethylated, thus restricted) DNA. This was established in our previous work (Gordeeva et al, *NAR*, 2019) where we have shown that lambda genomes are not restricted by BREX after induction from BREX+ lysogens, meaning that a round of replication inside a BREX+ strain is sufficient for subsequent protection, confirming an epigenetic mechanism. We have shown that this protection is gained upon m6A methylation of specific sites (BREX sites). In this work, we further show that methylation is introduced by BrxX protein in the context of a larger BREX complex (BrxXCBZ), and that BrxX loses affinity to methylated DNA (as m6A clashes with the methyl group of SAM) providing a clear explanation for BrxX affinity to unmethylated vs methylated sites. Mutations in BrxX substrate-binding pocket abolish protection by BREX system. It follows that BrxX is a target recognition protein in BREX system that brings the latter towards unmethylated BREX sites as they naturally occur indeed in both host and foreign DNA.

At this stage we do not know what happens later when BREX complex must ‘choose’ between methylation and restriction activities: this is a much more difficult question requiring reconstitution of the whole BREX complex which is far beyond the scope of this study. Logic, and comparison with other R-M systems suggests that BREX complex should protect hemi-methylated (host) DNA, while restricting unmethylated (foreign) DNA; this might require interactions between two BREX complexes bound in opposite orientations to two BREX sites.

Let us consider a hypothetical R-M immune system that is supposed to discriminate between phage and host DNA by detecting methyl marks on DNA. One clear difference allowing to distinguish them is that phage DNA completely lacks methylation, while replicated host DNA retains methylation on the template DNA strand and lacks methylation on the newly synthesized strand. Thus, a detection of a single unmethylated BREX site does not yet allow to discriminate between the host and the phage DNA: a system activated in this way would have inevitably targeted all newly replicated host DNA. Hence, we postulate that at least two BREX complexes are required, bound to two unmethylated BREX sites on the opposite DNA strands, that might interact with each other to make a collective ‘decision’ about restriction activation. As this is a purely speculative (although plausible) model at the moment, we only mention it in a discussion section. Answering this question experimentally, as well as closely related question of how methylation activity is controlled, is a subject of future and ongoing studies in our laboratories. Likewise, understanding the ‘exact defense mechanism of BREX’ may well take one’s entire academic career rather than be a subject of a single paper.

To address Reviewer's comment, we introduced textual changes throughout the text to point more clearly on the advance achieved in this paper and highlight the gap in understanding that still exists. The newly added text is marked in yellow.

2. The authors constructed a series of BrxX mutants targeting key residues important for DNA binding, recognition, and Ocr binding and evaluated their effects. However, there are still many issues that need to be addressed before the manuscript can be considered for publication:

--a. In Table S1, the authors refer to three loop mutations: 593DGAT599, 588EYSGFDGATVPI602, and 597VTA599. However, the authors did not clarify how these loops were mutated - were they deleted or substituted with other residues? This should be explained.

We aimed to substitute the MTD loop responsible for the recognition of the fifth position in BREX site of *E. coli* HS BrxX with the residues forming analogous loop in *E. fergusonii* BrxX. Based on the structural modeling, 3 variants of the substitutions were constructed: 593(GQISGEV)599 to 593(DGAT)599, 588(GARAFGQISGEVVQT)602 to 588(EYSGFDGATVPI)602, and 597(GEV)599 to 597(VTA)599. Unfortunately, these variants lost BREX defense activity, therefore they have not been investigated further and were not discussed in the previous variant of the manuscript. We have extended this description now.

--b. For the mutations in key amino acids involved in DNA binding and BREX site recognition, the authors only evaluated the effect of one of the mutations on DNA binding. For the other mutations, only the effect on BREX defense was measured. However, the effect on BREX defense alone does not functionally validate that these residues are directly involved in DNA binding or site recognition, as the authors claim. It is possible that the loss of defense could simply be due to improper expression or folding of these mutants in bacteria.

The authors should evaluate the effect of these mutations on DNA binding or site-specific recognition *in vitro*, similar to the approach used for K515E/K522E/K526E/K533E. In addition, the effect of these mutations on Ocr binding should also be evaluated for the Ocr-binding mutants.

According to Reviewer's initial request, we constructed a series of mutants to probe all discussed interaction interfaces: non-specific DNA backbone binding, sequence-specific DNA bases recognition and Ocr binding. All mutants were first tested on their effect on BREX defense, and only a subset of mutants were selected for *in vitro* validation, since it required to carry on an additional round of molecular cloning. The effect of the substitution of positively charged residues mediating non-specific DNA binding was demonstrated by showing that the K515E/K522E/K526E/K533E quadruple mutant has decreased DNA affinity. Given the clear negative phenotype we decided not to investigate this further: we do not find it interesting to deliberately break biological systems as it is always so easy to do while not always insightful.

It was far more interesting for us to try to engineer (= understand better) DNA recognition specificity by tweaking the selected TRD residues according to predicted recognition mechanism, based on models for *Salmonella typhimurium* and *Escherichia fergusonii* proteins. We were able to dramatically change the recognition specificity without affecting DNA binding as we show in the manuscript. By doing this, we demonstrated that different residues are involved in non-specific vs specific DNA recognition. For Ocr, by comparing defense phenotype for wild type and Ocr- phages we were also able to clearly demonstrate specific role of selected residues in Ocr recognition. Further experiments quantifying binding of different mutants to Ocr are beyond the scope of this study (will require development of suitable protein-protein interaction assays) and in our view are not necessarily insightful.

3. Furthermore, there are many illogicalities in the article that make it difficult to understand this manuscript.

--a. The authors state: “We show that BrxX is the only BREX component that can recognize BREX sites in vitro. BrxX also binds to non-methylated BREX sites in phage DNA in vivo. Thus, BrxX must be involved in viral DNA sensing and the initiation of the BREX defense”(Line 620-623). This conclusion is difficult to follow because it only shows that BrxX is necessary for binding to BREX sites, but does not exclude the possibility that other mechanisms could initiate the BREX defense. In fact, Reviewer 1 raised a similar concern, and the authors' response in the rebuttal letter only suggests that BrxX is important for BREX site recognition and BREX defense. However, there is no evidence that BREX site recognition by BrxX is temporally prior to BREX system activation, nor is there direct evidence that BrxX binding to BREX sites is a direct trigger for BREX system activation.

As already described above, BREX sites modification is both necessary and sufficient to provide protection against BREX (see Gordeeva 2019); current work shows that BrxX (acting as a part of BrxBCXZ complex) is solely responsible for this modification, and that binding of BrxX to BREX sites is required for BREX defense (as shown by Y511A result). We do not pretend to understand the mechanism beyond BrxX binding to unmethylated sites and following requests from other reviewers, decided to minimize speculations on how BREX system as a whole might work, and what molecular events control decision-making.

--b. The authors write: “The most widespread Type I BREX employs an R-M like principle: it modifies cell's own DNA at specific sites, and the acquisition of BREX-specific modification protects infecting phage from exclusion” (Line 61-63). This sentence is unclear - does the acquisition of BREX-specific modification refer to host DNA or phage DNA? The broad generalization of background knowledge in this single sentence may confuse readers unfamiliar with the BREX system. The authors should clarify which DNA is being modified and provide more context for readers.

We agree that this sentence was not clear, and sorry if it caused confusion. Changed to:

“The most widespread Type I BREX employs an R-M like principle: it modifies cell's own DNA at specific sites (BREX sites), and this epigenetic mark protects host DNA from BREX activity. Invading foreign DNA does not have the modification and is sensitive to BREX defense. The acquisition of BREX-specific modification is sufficient to protect infecting phage from exclusion.”

4. The authors seem to have overlooked my earlier advice to include more information in the figures. The current figures still have problems that distract the reader and contain several errors:

We have fully addressed specific requests that the Reviewer initially brought up. The points mentioned below were not included in the original evaluation but we thank the Reviewer for offering further guidance. We have added more information to figure legends where requested.

--a. The authors refer to 'a positively charged lining of the upper jaw' consisting of K226, K522, K515, and K533 of MTD (lines 291-292), citing Figures 2F and 3A. However, Figure 2F does not show the positions of these amino acids, and Figure 3A does not display all the mentioned residues.

There is a significant concentration of arginine and lysine residues around the region where DNA binds, forming a positively charged groove which is not unlike what is observed in other DNA binding proteins e.g. DNA topoisomerases. The purpose of Figure 2F is to demonstrate this positively charged surface, and labelling individual amino acids in our view would cause confusion without offering further conceptual insight into how BrxX interacts with DNA. We agree that it might be useful to have a separate reference to the positions of mentioned amino acids; the exact positions of DNA involved in the charged interactions is now included in **Supplementary Figure S5B**.

Figure 3A displays only interactions between BrxX and the BREX site six base pair sequence. We have adjusted the figure legend to make this clear. To show all interactions with DNA backbone was

never our intention and in our view would only cause confusion to the reader. We have adjusted the figure legend and manuscript text accordingly:

“Supplementary Figure S5... (B) Amino acid residues contributing to the positively charged upper and lower “jaws” of the BrxX monomer interacting with DNA backbone. “

“Figure 3A. (A) A scheme of interactions between BrxX and the six base pair BREX site. Distances in Angstrom are indicated. An asterisk indicates that the main chain and not the side chain is involved. Colors correspond to amino acids from MTD (pink) or TRD (yellow).”

--b. The authors state that 'In contrast, in *E. fergusonii* BrxX, this loop is markedly shorter and is thus able to accommodate the T:A base pair (see Figure 3C)' (lines 326-327). However, in Figure 3C, *E. fergusonii* BrxX is not shown with a T:A base pair.

Thank you for mentioning this. There is no structure of *E. fergusonii* BrxX protein bound to its cognate DNA so the statement is our speculation based on *E. coli* BrxX-DNA model. We have changed the main text and adjusted the legend in Figure 3 to ensure that this is clear.

“The importance of this observation is revealed by comparison with the BrxX homologs from Salmonella typhimurium and Escherichia fergusonii modelled by AlphaFold2. These BREX systems recognize, correspondingly, the GATCAG and GCTAAT sequences^{31,32,58}. The loop in S. typhimurium BrxX is predicted to be very similar to the one in E. coli BrxX, as expected given the identity of the terminal base. In contrast, in E. fergusonii BrxX the equivalent loop is markedly shorter (see Figure 3CD), likely to be able to accommodate the T:A base pair. We attempted to substitute E.coli BrxX loop with the one from E. fergusonii to manipulate the recognition specificity, however, three different hybrids we tested lost anti-phage defense (Supplementary Table S1).”

“Figure 3... (C) A species-specific MTD loop recognises the sixth base pair. A structural superposition is shown of loop 587-601 of E. coli HS BrxX MTD (rose) and equivalent loops in Salmonella typhimurium (purple) and Escherichia fergusonii (magenta) based on AlphaFold DB models AF-A0A5A8QD96-F1 and AF-B7L3T0-F1 respectively. (D) Side-by-side comparison of above loops with amino acids predicted to be important for the recognition of the sixth base pair shown as sticks. Recognition sites for all three BREX systems are shown, and the last base pair of the recognition site is boxed (G6 for E. coli). As the cognate DNA structure is not available for S. typhimurium and E. fergusonii, sixth base pair of E.coli HS BrxX is shown for all three models. In case of E. fergusonii, a larger purine base would be replaced by a smaller pyrimidine, in line with an observed change in the MTD loop structure.”

--c. In Figure 6A, what is represented in green—DNA or SAM?

We agree that two shades of green may cause confusion. SAM is now represented in cyan throughout the manuscript and colored by heteroatom. The BREX site DNA is represented in green, and the rest of DNA gray. The figure has been adjusted as well as the figure legend to clarify this:

“Figure 6... (A) Surface representation of the BrxX monomer shown in gray bound to DNA (black) containing the BREX binding site (green). Inset highlights the only external access point to the SAM molecule enclosed within the catalytic pocket (cyan backbone, colored by heteroatom).”

--d. The authors mention T82 in Figure 5F (line 420), but there is no T82 in Figure 5F. Should this reference be to T821 instead?

We thank the Reviewer for finding this typo. This is correct, the reference should be to T821.

--e. What do the labels 90i, 60i, and 147.2i in Figure 1 mean?

We suppose it could be a file conversion issue, in the initially submitted document, we have labels 90°, 60°, and 147.2° labels in **Figure 2**. We checked the correctness of all labels on the re-uploaded figures.